health and disease and epidemiology

climate, temperature, seasonality, epidemic growth, second plague pandemic, *Yersinia pestis*

**Author for correspondence:**
Fabienne Krauer
e-mail: fabienne.krauer@lshtm.ac.uk

# The influence of temperature on the seasonality of historical plague outbreaks

Fabienne Krauer[1,2], Hildegunn Viljugrein[1,3] and Katharine R. Dean[3]

[1]Centre for Ecological and Evolutionary Synthesis CEES, University of Oslo, Norway
[2]Centre for Mathematical Modelling of Infectious Diseases CMMID, London School of Hygiene and Tropical Medicine, London WC1E 7HT, UK
[3]Norwegian Veterinary Institute, Ås, Norway

 FK, 0000-0001-8341-4011; HV, 0000-0002-3798-5267; KRD, 0000-0003-2262-0385

Modern plague outbreaks exhibit a distinct seasonal pattern. By contrast, the seasonality of historical outbreaks and its drivers has not been studied systematically. Here, we investigate the seasonal pattern, the epidemic peak timing and growth rates, and the association with latitude, temperature, and precipitation using a large, novel dataset of plague- and all-cause mortality during the Second Pandemic in Europe and the Mediterranean. We show that epidemic peak timing followed a latitudinal gradient, with mean annual temperature negatively associated with peak timing. Based on modern temperature data, the predicted epidemic growth of all outbreaks was positive between 11.7°C and 21.5°C with a maximum around 17.3°C. Hence, our study provides evidence that the growth of plague epidemics across the whole study region depended on similar absolute temperature thresholds. Here, we present a systematic analysis of the seasonality of historical plague in the Northern Hemisphere, and we show consistent evidence for a temperature-related process influencing the epidemic peak timing and growth rates of plague epidemics.

## 1. Introduction

Seasonality is a salient feature of many infectious diseases [1]. For example, in the Northern Hemisphere, influenza is known to occur almost exclusively in winter, chickenpox peaks mainly in spring, West Nile Virus circulates mostly in summer [2]. The drivers of seasonality are manifold and depend on the type of transmission. Directly transmitted childhood diseases like measles are strongly affected by seasonal changes in social behaviour like school terms or household clustering. The seasonality of vector-borne diseases depends heavily on the seasonal abundance of the vector and additional hosts, which is a function of local climate including temperature and precipitation. The seasonality of respiratory diseases is influenced by climatic factors that affect the transmissibility of the pathogen, the susceptibility, and the contact patterns of humans [3]. Influenza and respiratory syncytial virus both show a clear latitudinal gradient with peak activity being shifted towards the end of the year for increasing latitudes and decreasing average temperatures [4].

Modern plague is another example of a highly seasonal disease. This zoonosis is maintained in rodent reservoirs and is transmitted between the animals through various flea species. Recurrent epizootic outbreaks may occasionally lead to spill over to the human population. Humans develop bubonic plague when bitten by infectious rodent fleas, which is usually fatal in about 66% of the cases if untreated [5]. They may also develop secondary pneumonic plague if the disease spreads to the lungs. Direct human-to-human transmission through infectious droplets leads to primary pneumonic plague, which is almost always fatal if not treated immediately. Many of the biological components of the plague system—including the bacteria, vectors, and animal hosts—are sensitive to environmental factors [6,7]. Temperature ranges between 15 and 27°C have

been associated with plague incidence in many African countries [8]. A recent study has shown that in India during 1898–1949, outbreaks were more likely to occur when the annual humidity was moderate (60–80%) compared to lower or higher humidity levels [9]. The sensitivity of the vectors and hosts to meteorological conditions often leads to a typical seasonal plague incidence in humans, which varies by geographical location (see e.g. [10]).

Contrary to the rich evidence for the seasonal behaviour of modern-day plague, the seasonality of plague during the Second Pandemic (14th to 19th century) has not been studied systematically. Anecdotal evidence suggests that historical outbreaks in Europe may have also followed a typical seasonal pattern. Parish records in Switzerland show that mortality during plague years was highest in November [11]. In the British Isles, plague season was predominantly in autumn [12], whereas in the Mediterranean region plague outbreaks peaked mainly in spring to summer [13]. Some have argued that historical plague in Europe peaked in the 'warm season' whereas modern plague in Asia peaks in the 'cool season' and speculated these discrepancies may be due to different causative agents [14]. However, molecular methods have shown unequivocally that Second and Third Pandemic plague was indeed the same disease caused by *Yersinia pestis* [15]. Nevertheless, the question of whether there was a consistent plague season, and how it related to meteorological conditions is still unaddressed. Quantifying the seasonality of an infectious disease and its drivers may also provide insight into the mechanisms of historical plague in Europe.

Thus, we here investigate the seasonality of plague outbreaks across Europe and the Mediterranean during the second plague pandemic. We collected and digitized epidemiological curves from published literature and established whether a seasonal pattern was present in the data. We then estimated the peak timing and the growth rates and studied how these measures related to temperature and precipitation. We show that temperature could partially explain both the epidemic peak timing and the concurrent epidemic growth rate. We conclude our analysis with a discussion about potential mechanisms and limitations of our work.

## 2. Methods

### (a) Plague and all-cause mortality data

We first established a dataset of plague outbreaks through a combination of systematic and opportunistic literature searches. We searched three major literature databases (jstor, PubMed, and the Internet archive) using a pre-specified search strategy (electronic supplementary material, table S1). The details are given in electronic supplementary material, text S1. We collected time-series data of incident plague and all-cause mortality during plague outbreaks at daily, weekly, bi-weekly, or monthly time intervals. We focused on pre-industrial outbreaks during the second plague pandemic (14th–19th century) in Europe (including Russia) and the Mediterranean Region. An epidemic was defined as uninterrupted plague activity for a period of at least three months for a given place. Epidemics with zero cases for four or more months in between were considered two different epidemics and split into two datasets. Some datasets had continuous plague activity during multiple years (recurrent peaks). For the calculation of the epidemic peaks, we split these epidemics into annual outbreaks at the minima between the peak to assess the seasonality for each outbreak separately. We also calculated

the estimated attack rates (i.e. the proportion of the population infected at the end of the epidemic, also known as final size) of all outbreaks, for which we could retrieve initial population sizes. The attack rates were calculated as

$$\text{attack rate} = \frac{n_{\text{deaths}} * \dfrac{1}{p(\text{dying} \mid \text{infected})}}{\text{population size}}, \tag{2.1}$$

where $n_{\text{deaths}}$ is the cumulative number of deaths at the end of the epidemic and $p(\text{dying} \mid \text{infected})$ is the case fatality ratio (CFR), which is around 0.66 in untreated cases [5]. Wherever possible, dates were converted to the Gregorian calendar prior to analysis. The raw data and the R code for data preparation and analysis are available in a public repository [16].

### (b) Climatic, geographical, and demographic data

Geographical coordinates of all locations with plague data were obtained from Google Maps. To investigate the effect of temperature and precipitation, we used the University of East Anglia Climatic Research Unit (CRU) TS 4.03 dataset, which are modern monthly averaged climate raster data [17]. From this dataset, we extracted the monthly temperature and precipitation values for the years 1901 to 1939. We first calculated the annual mean precipitation and temperature for each place and used these point estimates as predictors to model the timing of the epidemic peak. We then used the monthly temperature data and fitted a Fourier series model of the form

$$\begin{aligned}
\text{temperature} = a &+ b \times \sin\frac{2\pi}{365}t + c \times \cos\frac{2\pi}{365}t + d \\
&\times \sin\frac{4\pi}{365}t + e \times \cos\frac{4\pi}{365}t,
\end{aligned} \tag{2.2}$$

where $t$ is the day of the year. This model allows for a harmonic smoothing of temperature data without the constraints of vertical symmetry as in a simple sine function. We used the resulting coefficients to predict the average, daily temperatures, which were used as a covariate to model the growth rates.

### (c) Detection of seasonal pattern

We first investigated whether seasonality was present in our dataset. For this, we selected all plague-specific mortality data from places with at least four outbreaks. We first examined the distribution of the number of deaths by month for each place separately. We then grouped months into seasons and fitted a generalized linear model with a negative binomial error structure and a log link function and with the season as a categorical covariate for each place:

$$\ln\left(\frac{y}{\text{population}}\right) = \beta_0 + \beta_1 \cdot \text{season}, \tag{2.3}$$

where $y$ is the number of deaths, the natural log of the population is an offset, and $\beta_0$ is the intercept. The exponential of the $\beta_1$ coefficients resulting from this type of model give an estimation of the incidence rate ratio (IRR), which is the increase in incidence compared to a baseline (winter in our model). This model was then compared to a null model with an intercept only and without season by means of a likelihood ratio (LR) test.

### (d) Influence of latitude, temperature, and precipitation on the epidemic peak timing

To investigate the association between latitude and plague mortality, we aggregated all plague mortality data monthly and

calculated the annual average percentage (AAP) for each month $i$ as

$$AAP_i = \frac{n_i}{\sum_1^{12} n_i} \times 100\%, \tag{2.4}$$

where $n_i$ is the number of plague deaths in month $i$ and examined the distributions with a plot. Since latitude itself is a proxy measure for climatic variables, we then investigated the influence of annual average temperature and precipitation on the observed patterns of mortality. For this, we looked at the timing of the mortality peak, which is a more convenient indicator for seasonality of historical outbreaks than the start or end points, which are often not well recorded. We aggregated all daily and weekly plague mortality data by International Organization for Standardization (ISO) week and calculated the calendar week in which most deaths were reported. Some places had multiple outbreaks, and these data points are potentially more strongly correlated than measurements from other places. We thus fitted two linear, univariable GEE (generalized estimating equations) models for the effect of temperature and precipitation, respectively, using the R package geepack [18]. The GEE model is a population-averaged model that can deal with the clustering of observations [19]. It treats the within-subject covariance as a nuisance effect but yields a parameter estimate and robust standard errors adjusted for the clustering. We assumed that the correlation of observations within the same cluster was constant across all clusters. For each model, we calculated the variance explained given by the coefficient of determination adapted to GEE models ($R^2_{marg}$, for the calculation see [20]).

## (e) Influence of temperature on epidemic growth

Based on the findings from the model of temperature and peak timing, we were interested in how temperature determined epidemic growth. To calculate the growth rates we used all weekly plague mortality data with complete epidemiological curves. We then established the number of incident infections over time by taking into account the probability of dying when infected and the delay between infection and death. For the prior, we divided the weekly incident deaths by the average CFR. For the latter, we subtracted the average time from infection to death from all the dates, which is around seven days according to data from the third plague pandemic in India [21]. We then modelled the epidemic growth rate for each outbreak by regressing the log-transformed incident cases on time in days. To account for variations of the growth rate over time, we partitioned the data into rolling time windows of four weeks and fitted each segment separately. This resulted in a time-varying, moving average of the growth rate. To reduce the influence of random fluctuations when case numbers were low, we excluded the beginning and the tail of the epidemics when the incident number of cases was smaller than 21. To explore the potential function that links temperature to the time-varying growth rates ($r$), we fitted a univariable generalized additive mixed model (GAMM) with a Gaussian error structure and an identity link function of the form:

$$r = \beta_0 + f(\text{temperature}) + \varepsilon, \tag{2.5}$$

where $\beta_0$ is the intercept and $\varepsilon$ are the residuals. To account for the remaining autocorrelation of data points from the same outbreak, we assumed an AR1 Gaussian correlation structure for the residuals with $\varepsilon_i = \varepsilon_{i-1} + \rho e_i$, where $\rho$ is the correlation parameter and $e_i \sim N(0, \sigma^2)$. The advantage of a generalized additive model is that it allows for a nonlinear relationship between predictor and outcome without making an explicit a priori assumption about the shape of this relationship (see [22]). We used the R package mgcv for the fitting procedure [22].

# 3. Results

## (a) Description of plague and all-cause mortality data

With the systematic search strategy, we retrieved 2450 references (electronic supplementary material, table S2). After screening the full texts of 246 references (10%), we included nine publications that contained useful data. We also included 36 publications from an opportunistic search strategy resulting in 45 publications [11–13,23–63] containing useful data for our analysis. Eighteen publications contained data on two or more epidemics, which resulted in 130 datasets in total, with 76 datasets for plague mortality and 54 datasets for all-cause mortality during plague outbreaks (electronic supplementary material, tables S3 and S4). Some of the epidemics consisted of multiple years resulting in a total of 157 annual outbreaks from 87 unique locations. Of these, 100 (64%) were plague mortality data and 57 (36%) were all-cause mortality data. The location and epidemiological curves of all outbreaks are shown in the electronic supplementary material, figures S1 and S2. The interval at which the plague deaths were recorded was weekly (28.6%), monthly (36.4%), daily (33.8%), or bi-weekly (1.3%). The majority of the plague mortality datasets were from the UK (35.1%), followed by Spain (18.2%), Egypt (10.4%), and other places (36.3%). Most dated from the 17th century (32.5%), followed by the 16th century (23.4%) and the 19th century (18.2%). The median duration of the plague outbreaks was nine months (range 4–103). The median cumulative number of deaths was 1352 (range 38–68 596) and the median number of deaths at peak was 429 in a month (range 14–31 036). The median attack rate was 11% (0.11, range 0.0009–1) and decayed nonlinearly with increasing initial population size (electronic supplementary material, figure S3). Above a population size of 100 000 the proportion of infected people never exceeded 0.25 (25%). The year of the outbreak was not associated with the attack rate (Pearson correlation coefficient −0.05).

## (b) Detection of seasonal pattern

As shown in figure 1, the monthly distribution of plague deaths in places with at least four outbreaks (Alexandria, Algiers, Barcelona, London, and Vienna) suggests a distinct seasonal pattern for all places but different peak times. A model including season explained the data better than a model without a season covariate for all places except Algiers (electronic supplementary material, table S5, see $p$-values of LR tests). Compared to the baseline of winter, the maximum increase in the incidence was observed in spring (March–May) for Alexandria (IRR 4.75, 95% CI 1.7–12.54) and Barcelona (IRR 8.1, 95% CI 3.31–18.61), and in autumn (September–November) for London (IRR 10.66, 95% CI 6.85–16.57) and Vienna (IRR 5.11, 95% CI 2.03–12.55). In Algiers, the maximum incidence was also in spring but the difference to the other seasons was statistically not significant (IRR 2.66, 95% CI 0.82–8.47). These results indicate that the historical outbreaks in our data exhibit a consistent seasonality pattern.

## (c) Influence of latitude, temperature, and precipitation on the epidemic peak timing

The findings from the previous model not only suggest that historical plague showed a distinct seasonality, but also that the typical plague season differed by latitude. We

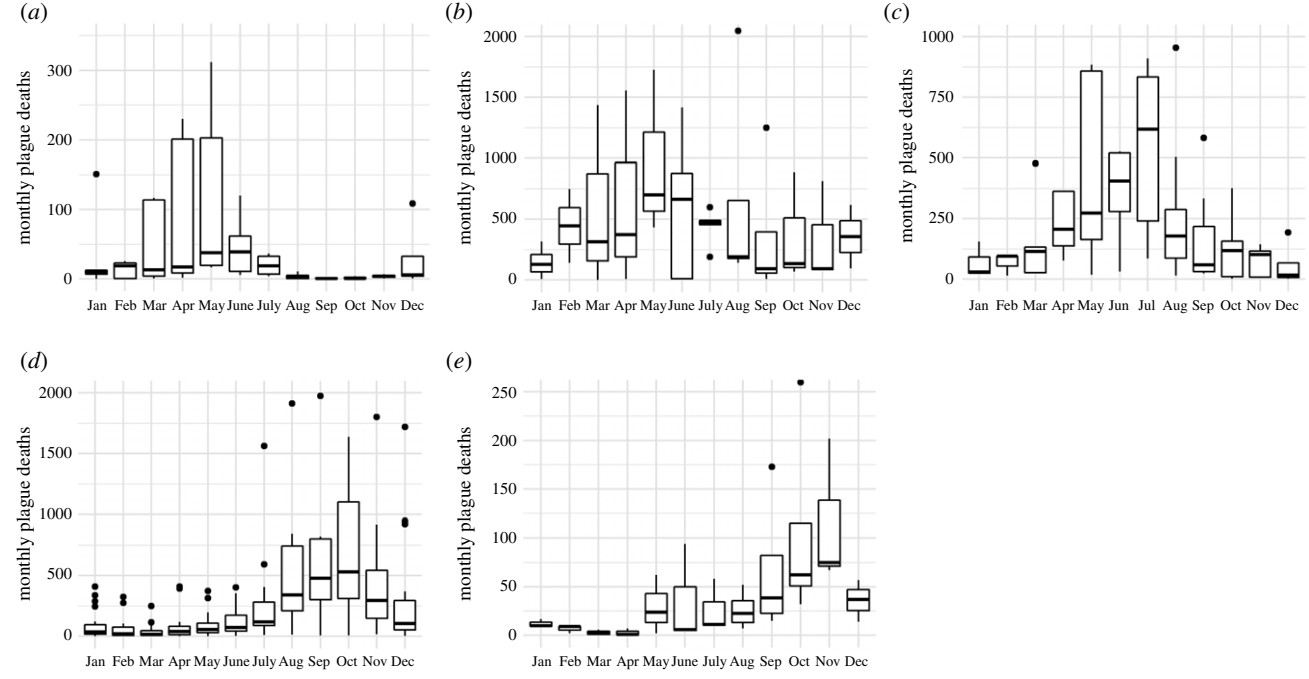

**Figure 1.** The distribution of plague deaths by months from all recorded plague years shows a distinct seasonality pattern for all five places ((a) Alexandria, (b) Algiers, (c) Barcelona, (d) London, and (e) Vienna) with multiple outbreaks. The boxes encompass the 25th to 75th percentiles, the whiskers extend to the maximum or to maximally 1.5 times the interquartile range. The vertical lines denote the medians. Few outliers (dots) have been omitted from the plot for clarity.

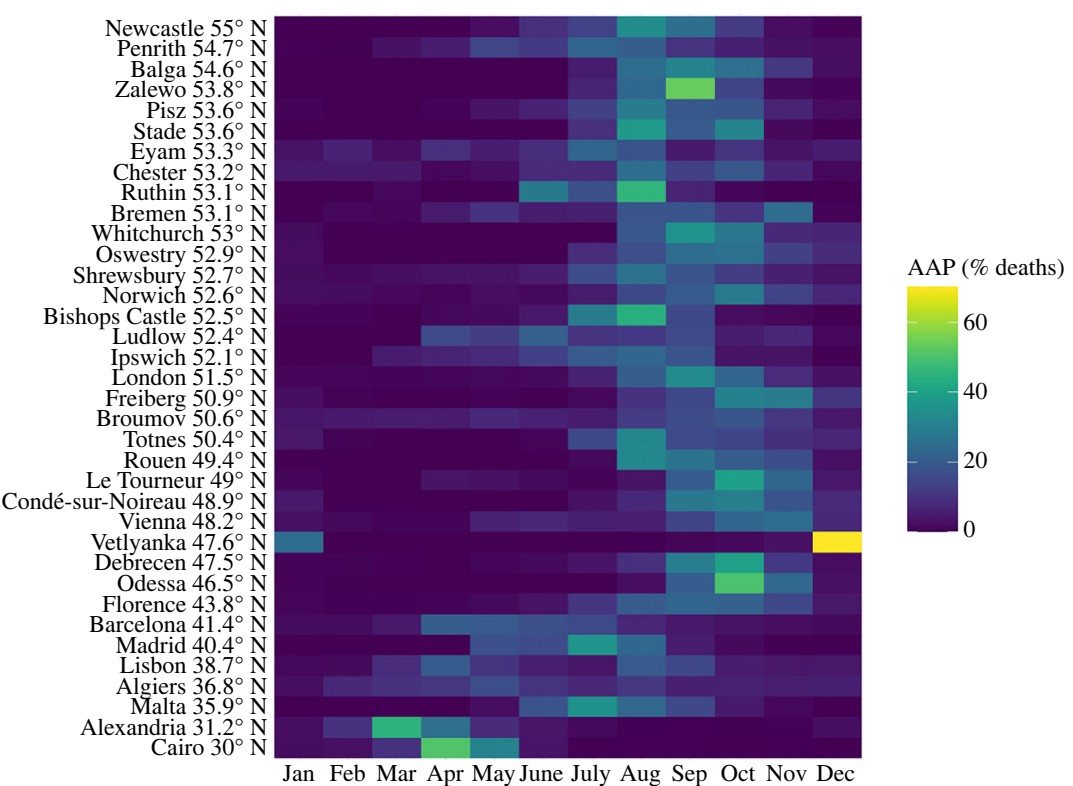

**Figure 2.** The AAP of monthly plague deaths suggests a latitudinal gradient with a shift of epidemic activity towards the end of the year for increasing latitude. (Online version in colour.)

therefore investigated this aspect further by assessing the monthly percentage distribution of plague deaths for all places in our dataset. Figure 2 suggests a latitudinal gradient with a shift of the cases towards the end of the year for increasing latitude.

Latitude alone may not explain the seasonality of plague mortality mechanistically, but it is a rough proxy measure for climatic conditions. We thus investigated the association between the seasonal epidemic peak timing and average, annual temperature and precipitation, respectively. For this, we used weekly plague mortality data from 65 outbreaks from 22 unique places, of which 73% had data for only one outbreak. A third of the outbreak data came from London. The median peak was observed at week 35 (range 11–52), which falls usually at the beginning of September (range mid-March to end December). Overall, we found a negative

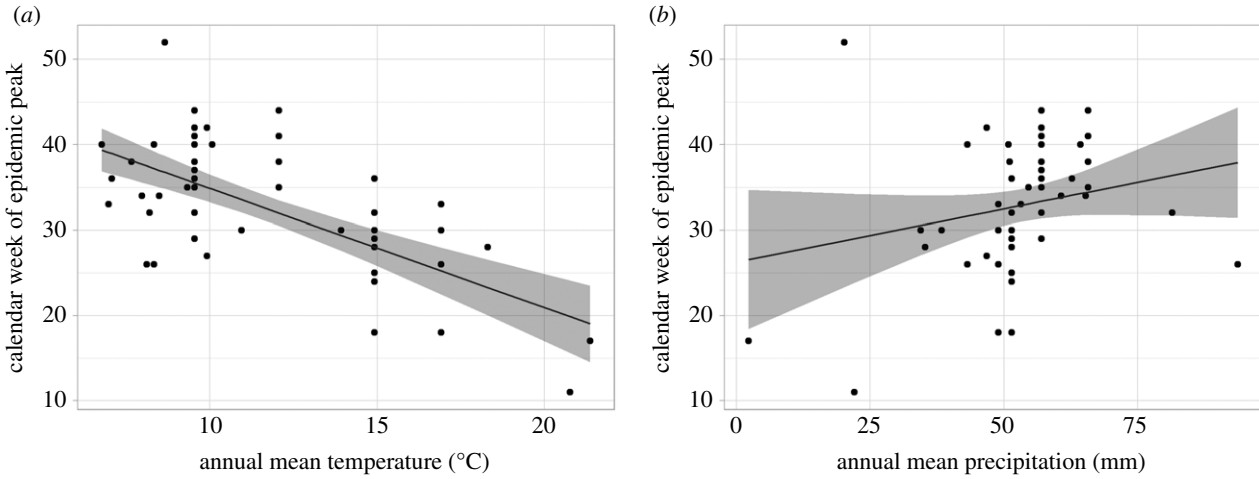

**Figure 3.** Association of annual mean temperature (*a*) and annual mean precipitation (*b*) with epidemic peak week. The bands show the 95% CIs for the fit from a univariable, linear GEE model.

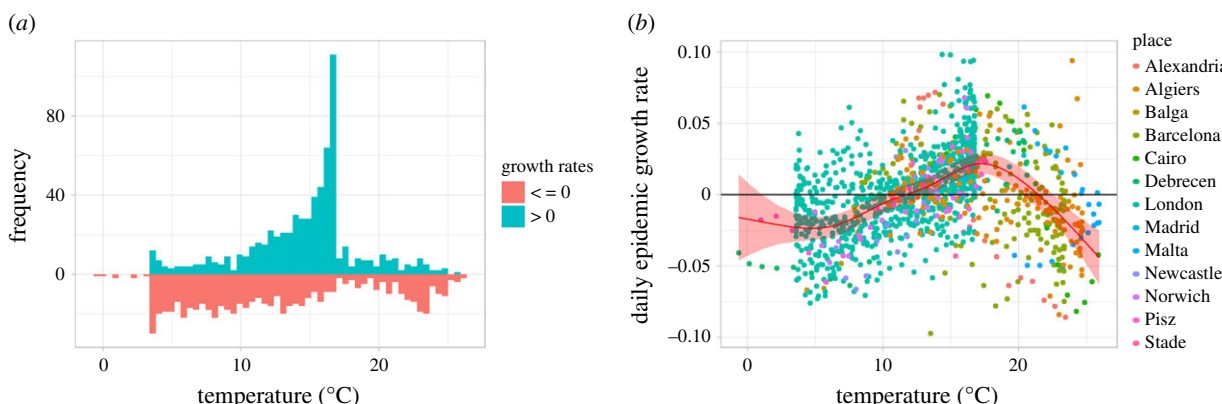

**Figure 4.** Distribution of predicted time-varying growth rates by temperature. (*a*) Histogram of positive (green) and zero or negative growth (red) and (*b*) scatter plot of time-varying growth rates for the full dataset. The red line and ribbon indicate the fit and 95% CI from a univariable GAMM model. The maximum growth was predicted at 17.3°C. (Online version in colour.)

association between week of the year and annual temperature, and no association between week and precipitation (figure 3). A decrease in average temperature of 1°C delayed the epidemic peak by 1.4 weeks ($\beta = -1.39$, s.e. = 0.232, $p < 0.01$). Temperature was a significant predictor in the model, but overall the model explained only 44% of the variance in the data ($R^2_{\mathrm{marg}} = 0.44$). Average annual precipitation was not associated with peak month ($\beta = 0.124$, s.e. = 0.130, $p = 0.34$, $R^2_{\mathrm{marg}} = 0.075$) and did not improve the fit when added to the model with temperature ($R^2_{\mathrm{marg}} = 0.43$).

To investigate the robustness of our findings, we repeated the analysis including additional data points. As shown in electronic supplementary material, figure S4, the negative association between temperature and peak timing was confirmed when extending the dataset with monthly plague data (beta = −0.31, s.e. = 0.03, $R^2_{\mathrm{marg}} = 0.497$), weekly all-cause mortality (beta = −1.33, s.e. = 0.177, $R^2_{\mathrm{marg}} = 0.429$), and monthly all-cause mortality (beta = −0.286, s.e. = 0.029, $R^2_{\mathrm{marg}} = 0.43$). Excluding smaller outbreaks with less than 100 deaths at the peak improved the model fit slightly (beta = −1.55, s.e. = 0.22, $R^2_{\mathrm{marg}} = 0.556$). These results corroborate the finding that annual mean temperature had an effect on the peak timing of historical plague epidemics in Europe and the Mediterranean: the epidemic activity

occurred later in the year as the mean temperature decreased and the latitude increased. Of note, the peaks represent the maximum mortality and not the maximum in epidemic activity. However, if we assume that the average time from infection to death was constant across all places and throughout the outbreaks, the epidemic peak is simply shifted to the left by a constant, which does not affect the results from the regression analysis.

### (d) Influence of temperature on epidemic growth

Based on the finding that peak timing was associated with annual temperature, we further investigated this aspect by examining the temperature dependence of the epidemic growth rates. For this, we used the plague mortality data from 49 outbreaks (13 unique places) and estimated the instantaneous time-varying growth rates for each outbreak separately (electronic supplementary material, figure S5). These calculated growth rates ranged from −0.097 to 0.098 with a median estimate of 0.02 for the increasing phases and −0.02 for the decreasing phases. We then explored the relationship of the instantaneous growth rates with the concurrent local temperature. The distribution of growth rates by temperature is noisy, but a nonlinear relationship with unequal distributions for positive and negative growth rates

was detectable (figure 4a). The temperatures measured at the 13 places ranged from −3.8°C to 27.2°C, but no positive growth was observed below 4°C or above 26°C. The prediction from a GAMM model showed that this relationship takes the form of a single-peak curve with a maximum at 17.3°C (figure 4b). Positive growth was predicted only for temperatures between 11.7°C (95% CI 9.8–13.4°C) and 21.5°C (95% CI 20.3–22.5°C). This model explained 28% of the variance in the data ($R^2_{adj} = 0.28$). The model fitted substantially better than a model without an AR1 correlated error structure (LR test p-value < 0.0001).

We also conducted two separate sensitivity analyses for the effect of temperature on the growth rates by (i) including only large outbreaks with at least 500 weekly deaths at peak and (ii) omitting the data from London. The prior addresses the issue of imprecise measurements of the growth rates in small outbreaks due to random fluctuations in cases, the latter addresses the predominance of London in the dataset. Using only large outbreaks, the nonlinear, unimodal relationship was still detectable (electronic supplementary material, figure S6A). The maximum growth was predicted marginally lower at 16.2°C, and the model fitted marginally better than with the full dataset ($R^2_{adj} = 0.30$). Fitting the model without the data from London still resulted in the same relationship between temperature and growth rates (electronic supplementary material, figure S6B) with a peak at 16.5°C. However, the variance explained was lower than in the model with the full dataset ($R^2_{adj} = 0.20$). We also performed the same analysis with precipitation as a predictor of the time-varying growth rate and found a weak, unimodal association ($R^2_{adj} = 0.19$) (electronic supplementary material, figure S7A). However, this association was driven entirely by the data from London: when we omitted these data points, there was no association between precipitation and the time-varying growth rate ($R^2_{adj} = 0.03$, electronic supplementary material, figure S7B). Model diagnostics showed that the residuals were approximately normally distributed and the variance was constant for all models.

## 4. Discussion

In this study, we show that the duration and the size of Second Pandemic outbreaks across Europe and the Mediterranean varied substantially and that the attack rates depended on the population size: the larger the initial population size, the lower the proportion of the population infected. This suggests that there was a limiting factor that prevented large populations from being infected completely, unlike small populations that could be wiped out almost completely. We further show that places with repeated outbreaks had a consistent plague season and that epidemic peak timing followed a latitudinal gradient. This same gradient was also observed when modelling the annual mean temperature as a predictor: the colder the mean temperature and the more north a place, the later the epidemic peak occurred. Moreover, we show that the predicted epidemic growth of all outbreaks was positive between 11.7°C and 21.5°C with a maximum at 17.3°C. These findings suggest that the growth of plague epidemics across the whole study region may have depended on similar absolute temperature thresholds. The minimum temperature threshold also explained the shift in peak timing with decreasing

temperature: the lower the annual mean temperature in place, the later in the year the threshold was reached (Pearson correlation coefficient −0.90, electronic supplementary material, figure S8) and thus the later in the year the epidemic peaks occurred. Overall, our findings from the different analyses provide consistent evidence for a temperature-related process that influenced the dynamics of pre-industrial plague outbreaks. Precipitation on the other hand did not seem to be associated with neither peak timing nor epidemic growth. Although we did not observe an effect of precipitation in our analysis, we cannot exclude that precipitation played a role in addition to temperature.

The use of modern meteorological data is a limitation of our study. As we show in the electronic supplementary material, historical temperatures may have differed from modern temperatures (electronic supplementary material, text S2 and figure S9). The Second Pandemic occurred during a prolonged period of cooling in Europe known as the Little Ice Age. Thus, the temperature range conducive of epidemic growth could have been marginally lower than our estimates. The availability of only averaged modern data also meant that we could not consider inter-annual differences in the seasonal temperature and precipitation, or their lagged effects. Our results should thus be considered an approximation of the general meteorological process at a seasonal scale. Contemporaneous meteorological data may have also explained more of the variance in the peak timing and growth rates, particularly for precipitation, which can exhibit large variation between years.

Our findings warrant a closer investigation into potential mechanisms. Plague is linked to climate through the sensitivity of vectors and hosts to meteorological conditions [6,7]. The mortality rates and development of various flea species are known to increase with rising temperatures leading to a characteristic seasonal abundance, but the favourable temperature ranges differ by flea species [64]. Moreover, the plague bacterium, Yersinia pestis, was shown to be transmitted from fleas to rodents at temperatures ranging from 23°C to 30°C but not at 10°C [65]. Ecological studies have shown that precipitation also influences plague incidence among humans through effects on vegetation, fleas, and hosts (trophic cascade hypothesis) often with a lag of some months to several years [7]. Today, plague transmission in humans happens mainly at the interface between the human population and wildlife, and its temperature dependence is thus primarily due to the temperature sensitivity of the wildlife plague system.

There is an ongoing debate about which plague hosts and vectors were present in Europe during the Second Pandemic and whether the disease was maintained in local reservoirs (see e.g. [66,67]). The establishment of a persistent reservoir in the Alpine region of Western Europe has been proposed [68]. However, recent studies of ancient DNA suggest that the genetic diversity of Y. pestis in Europe may be the result of multiple introductions [15] possibly from multiple reservoirs located in western Eurasia [69]. Since meteorological conditions can influence plague transmission through several pathways, the association in our study between temperature and the seasonal development of plague epidemics alone cannot provide evidence for or against a rodent reservoir in Europe. It is also unclear to what extent direct transmission routes (pneumonic or through human ectoparasites) played a role [67]. The contribution of climatic conditions to

pneumonic transmission has not been addressed systematically. However, the occurrence of pneumonic outbreaks in many climatically diverse places such as Manchuria in December to January ([70], average temperature around –10°C) or Rangoon (Myanmar) in September ([71], average temperature 25–30°C) suggests that the suitable temperature range may be very large, and that other factors such as crowding and the influx of susceptibles may be more influential. The influence of temperature on human ectoparasites and the potential effect on plague in humans has not been studied so far and thus, we cannot evaluate it in comparison to our results.

The occurrence of plague outbreaks in Europe during the Second and Third Pandemics has also been linked to maritime trade activities to explain outbreaks in the absence of permanent local plague reservoirs [66]. Ship trade in pre-industrial times was highly seasonal due to weather conditions and the availability of commodities. However, favourable local conditions are a prerequisite for an outbreak to grow beyond a few dozen cases after disease introduction (irrespective of the assumption about the transmission routes). Most of the data in our analysis are from large epidemics with hundreds of casualties, thus conditions must have been suitable for transmission. In fact, during the Third Pandemic, only around 5% of disease introductions in Europe through maritime trade activities led to the establishment of a sizeable outbreak with more than 20 cases (see [72]). This suggests that the local conditions did not favour epidemic growth or that mitigation strategies were very successful. Other factors such as the fluctuation of susceptibles due to population immunity and demographic turnover of the human population are also known to influence the periodicity of some infectious diseases. This periodicity arises mostly in directly transmitted childhood diseases, where the interplay of population immunity and replenishment of susceptibles through births leads to an endemic-epidemic cycle [73]. However, plague is a highly fatal disease and many of these outbreaks were several years to decades apart. The increase in immune hosts over time may have been counteracted by the rapid natural population turnover due to a short lifespan and a high birth rate. Thus, we expect that population immunity played only a minor role in the periodicity of most of these outbreaks.

Data from outbreaks not included in our study confirm our findings. The last large outbreak in Porto started in June 1899, but the incidence increased rapidly only in the second half of September with a peak in mid-October [74]. The average temperature dropped below 20°C in mid-September when the cases started to increase rapidly. A recently published study of a large epidemic in Venice in 1630/31 [75] showed that the peak mortality occurred approximately at the end of October (around calendar week 44), which is within the range of our data for places with a comparable annual mean temperature (12.6°C). The largest expected epidemic growth seems to have occurred at the beginning of September, when the predicted temperature decreased below 20°C (according to the CRU data), which also agrees with our model prediction. In the Malagasy highlands, outbreaks in the last decades commonly grow rapidly in August and September [76] when the average temperature is around 15–20°C, which is in the same range as predicted by our model. However, not all plague outbreaks may show the same temperature thresholds we found in our analysis. Outbreaks in Bombay usually increased in December to January and peaked in February to April [77]. The average temperature in Bombay drops below 25°C in December and January and rises again above 25°C in March. Similarly, in Uganda, outbreaks increase typically in August to October during a period of high precipitation when the temperature is around 22°C [78].

As with modern plague outbreaks, we argue that local, climatic conditions that acted on the abundance of vectors (and other hosts) were drivers of historical plague seasonality. Further entomological and epidemiological research is needed to investigate the influence of temperature and other meteorological factors on the abundance of various plague vectors and hosts, and the downstream effects on human plague mortality. Uncovering the underlying mechanisms of the plague seasonality that we document here will strengthen our understanding of historical plague transmission.

Data accessibility. The R code and the plague and all-cause mortality dataset are available from a public repository: http://doi.org/10.5281/zenodo.4240204 [16]. The CRU TS 4.03 dataset [17] is available at https://catalogue.ceda.ac.uk/uuid/10d3e3640f004c578403419aac167d82.

Authors' contributions. F.K.: conceptualization, data curation, formal analysis, investigation, methodology, validation, visualization, writing-original draft, writing-review, and editing; H.V.: supervision, writing-review and editing; K.R.D.: data curation, writing-review, and editing

All authors gave final approval for publication and agreed to be held accountable for the work performed therein.

Competing interests. We declare we have no competing interests.

Funding. This work was supported by funding from the Centre for Ecological and Evolutionary Synthesis (CEES), University of Oslo and the Research Council of Norway (FRIMEDBIO project 288551).

Acknowledgements. We thank everyone who helped with finding data or who contributed their own data: the staff at the Science Library (Realfagsbiblioteket) of the University of Oslo, Krásenská Klára (National Library of the Czech Republic), Dr Daniel Curtis (University of Rotterdam), Filomena Rodrigues (Associação Portuguesa de Demografia), Marie Prusova (Czech Statistical Office).

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
