## [Peer Review File · Proceedings of the Royal Society B: Biological Sciences]

Review History

RSPB-2020-1407.R0 (Original submission)

Review form: Reviewer 1

Recommendation

Major revision is needed (please make suggestions in comments)

Scientific importance: Is the manuscript an original and important contribution to its field?

Good

General interest: Is the paper of sufficient general interest?

Good

Quality of the paper: Is the overall quality of the paper suitable?

Acceptable

Is the length of the paper justified?

Yes

Should the paper be seen by a specialist statistical reviewer?

No

Do you have any concerns about statistical analyses in this paper? If so, please specify them explicitly in your report.

No

It is a condition of publication that authors make their supporting data, code and materials available - either as supplementary material or hosted in an external repository. Please rate, if applicable, the supporting data on the following criteria.

Is it accessible?

Yes

Is it clear?

Yes

Is it adequate?

Yes

Do you have any ethical concerns with this paper?

No

Comments to the Author

This is an interesting study trying to document the seasonality of historic plague epidemics in Europe and Northern Africa. This work has the potential to add to our understanding of historic plague outbreaks, but the findings have to be discussed in context. The overall scientific argument has to be strengthened.

Major comments:

The abstract does not reflect the rest of the document well and should be rewritten to include justification of the study and implications of findings.

Please strengthen the justification for your study throughout your document i.e. the importance to understand historic epidemics to prepare for future epidemics. That something simply has not been done before is not enough to validate a study.

A major concern is the lack of background information on historic plague outbreaks in pre-industrial Europe. The authors need to provide a stronger argument for their hypothesis and a more nuanced discussion of their findings and the implications of their work. Literature suggests that the main driver for plague outbreaks in pre-industrial Europe and North-Africa was the maritime trading system (1). Indeed, the majority of the data on plague mortality records come from either port cities themselves, or from cities linked to the historic maritime trading system. Yet, the authors make a case for the seasonality of vector borne diseases such as plague mainly due to seasonal abundance of host and vectors. In the methods section the authors mention "introduction events" for the first time, but it is not clear if they refer to the absence of a reservoir or the zoonotic event. Indeed, studies suggest that there is no evidence for a plague reservoir in pre-industrial Europe (2), reducing the strength of a link between season, host and vector availability and plague even further. This may be the reason why model fit was so poor.

However, linking the dynamics of plague outbreaks (growth rates) to climate variables, as has been done here, is sensible and in my opinion the findings of this part should be predominantly discussed. Not the occurrence of an outbreak, but its persistence and growth may be linked to environmental factors such as temperature, as it depends on the vector and host availability after the introductory event. The results on growth rate should be discussed in the context of a (temperature-modulated) trophic-cascade model for plague, including climatic effects on rodent abundance, flea abundance, and pathogen transmission.

It is not surprising that using average annual temperature and precipitation does not give clear results, as the plague transmission system relies on specific temperature and humidity ranges.

Even though the lack of good data is mentioned in the limitations, the impact is not.

The discussion seems rushed and lacks clarity. The explanations of the studies limitations take up most of the space. It would improve flow if the argument was strengthened. Priority should be given to the findings on growth rates and their link to seasonality and temperature ranges. The

difference between pneumonic and bubonic plague outbreaks in terms of sensitivity of transmission pathways to temperature should be clarified. Second pandemic outbreaks in Europe and northern Africa have been linked to multiple re-introductions via trade routes. Good climate conditions for vectors and hosts can positively affect growth rates after introduction.

Minor comments:

1. 22 Please add "in the northern hemisphere".
1. 41 please add: ..April and have been linked to temperature and precipitation (3, 4)
1. 48 briefly describe the difference between modern day outbreaks and historic outbreaks in terms of transmission mechanisms. Endemic plague vs. introductory events on trade routes (1).
1. 140 It is not clear why you used monthly plague mortality AND all-cause mortality in the same model? I assume you did not use both for the same location and outbreak.
1. 218 please enter the exact p-values for the LR test until <0.001
1. 222 change to "from 22 unique places"
- 1.228 indicate the direction of the 1.4 week shift
- Fig 2. I suggest considering having calendar week on the x-axis. Here it can be seen that precipitation is not linearly related to epidemic calendar week and that should be added to the limitations of the study.
1. 352-358 The outbreak in Madagascar was mainly pneumonic as mentioned (5), strongly linked to human behaviour. The authors should add that most bubonic cases were recorded from the beginning of the usual plague season in Madagascar at the beginning of October when the climatic conditions are most favourable for vectors (6).

References

1. Yue RPH, Lee HF, Wu CYH. Trade routes and plague transmission in pre-industrial Europe. *Scientific Reports*. 2017;7(1):12973.
2. Schmid BV, Büntgen U, Easterday WR, Ginzler C, Walløe L, Bramanti B, et al. Climate-driven introduction of the Black Death and successive plague reintroductions into Europe. *Proceedings of the National Academy of Sciences*. 2015;112(10):3020.
3. Kreppel KS, Caminade C, Telfer S, Rajerison M, Rahalison L, Morse A, et al. A Non-Stationary Relationship between Global Climate Phenomena and Human Plague Incidence in Madagascar. *PLoS Negl Trop Dis*. 2014;8(10):e3155.
4. Stenseth N, Samia N, Viljugrein H, Kausrud K, Begon M, Davis S, et al. Plague dynamics are driven by climate variation. *Proc Natl Acad Sci USA*. 2006;103(35):13110 - 5.
5. Tsuzuki SaL, Hyojung and Miura, Fuminari and Chan, Yat Hin and Jung, Sung-mok and Akhmetzhanov, Andrei R and Nishiura, Hiroshi, . Dynamics of the pneumonic plague epidemic in Madagascar, August to October 2017. *Eurosurveillance*. 2017;22.
6. Kreppel KS, Telfer S, Rajerison M, Morse A, Baylis M. Effect of temperature and relative humidity on the development times and survival of *Synopsyllus fonquerniei* and *Xenopsylla cheopis*, the flea vectors of plague in Madagascar. *Parasites & Vectors*. 2016;9(1):82.

Review form: Reviewer 2

Recommendation

Reject – article is scientifically unsound

Scientific importance: Is the manuscript an original and important contribution to its field?

Marginal

General interest: Is the paper of sufficient general interest?

Poor

Quality of the paper: Is the overall quality of the paper suitable?

Marginal

Is the length of the paper justified?

Yes

Should the paper be seen by a specialist statistical reviewer?

Yes

Do you have any concerns about statistical analyses in this paper? If so, please specify them explicitly in your report.

Yes

It is a condition of publication that authors make their supporting data, code and materials available - either as supplementary material or hosted in an external repository. Please rate, if applicable, the supporting data on the following criteria.

Is it accessible?

Yes

Is it clear?

Yes

Is it adequate?

Yes

Do you have any ethical concerns with this paper?

No

Comments to the Author

In their manuscript, Krauer et al. construct a large, historical data set on plague outbreaks and relate reported mortality to climate/weather variables. The authors have compiled and digitized an impressive set of data, which will undoubtedly facilitate quite a bit of future work on historical plague outbreaks. I also agree with the authors that studying the drivers of seasonality for infectious disease outbreaks remains an important and active area of research. While I found the seasonal differences between Algiers and Europe (including London)/Alexandria intriguing, given the biased geographic nature of these data and the complex role precipitation and temperature play in the transmission of other pathogens, I remain skeptical of the conclusions regarding the specific role weather/climate plays in modulating transmission. Indeed, as a detail below, there seem to be many alternative hypotheses for how plague outbreaks are structured. I have the following, more specific comments, which I hope the authors find constructive.

1.) Do we have any sense for how these data might be biased due to general patterns of under-reporting and/or differences due to reporting practices by country? It may be that the signs of plague death are sufficiently telling that reporting is less of an issue than for other diseases, but the authors should clarify this point.

2.) Related to the above, one concern about peak timing is that this may be heavily influenced by reporting. For example, Park and Bolker recently discussed these issues from a theoretical perspective.

Park, S. W., & Bolker, B. M. (2020). A note on observation processes in epidemic models. *Bulletin of Mathematical Biology*, 82(3), 1-8.

3.) Additionally, to my knowledge, most climate-forced infectious disease models focus on changes in the effective number of secondary infections, R_e . The peak time may or may not

change, depending on the population and household structure the disease is moving through, and other details of the assumed model. I would encourage the authors to focus on a different aspect of the epidemic curve or provide further justification for the use of peak timing..

Axelsen, J. B., Yaari, R., Grenfell, B. T., & Stone, L. (2014). Multiannual forecasting of seasonal influenza dynamics reveals climatic and evolutionary drivers. *Proceedings of the National Academy of Sciences*, 111(26), 9538-9542.

Volz, E. M., Miller, J. C., Galvani, A., & Meyers, L. A. (2011). Effects of heterogeneous and clustered contact patterns on infectious disease dynamics. *PLoS Comput Biol*, 7(6), e1002042.

4.) Can the authors expand a bit on how specifically their GEE model handles clustering and whether by clustering the authors mean having multiple observations from the same geographic location? It's more natural for me to think in terms of fixed effects from a mixed-effects model where location is a random effect or the central moment of the distribution describing a hyper parameter in a hierarchical Bayesian framework. I'm not necessarily advocating for a different model, just a bit more clarity.

5.) Did the authors consider leave-one-out cross validation performed at the country- or city-level where you both hold out individual outbreaks and entire locations? This might make a much more compelling case that there are general patterns related to temp/precip.

6.) Given all the myriad, climatic differences between these locations, e.g., Egypt, Algiers, Vienna, and London, how confident should we be that the associations with temp/precip aren't masking the true relationships? As an example, a paper on Whooping Cough outbreaks in London between 1700 and 1800 argued that climate variables were largely masking the effects of nutritional variation (or at least were only indirectly influencing disease dynamics).

Duncan, C. J., Duncan, S. R., & Scott, S. (1996). Whooping cough epidemics in London, 1701-1812: infection dynamics, seasonal forcing and the effects of malnutrition. *Proceedings of the Royal Society of London. Series B: Biological Sciences*, 263(1369), 445-450.

7.) For example, Ben Ari et al. 2011 argued that scale matters immensely when considering climate variables and modern plague dynamics.

Ari, T. B., Neerinx, S., Gage, K. L., Kreppel, K., LaDisoit, A., Leirs, H., & Stenseth, N. C. (2011). Plague and climate: scales matter. *PLoS Pathog*, 7(9), e1002160.

8.) Similarly, Cavanaugh & Marshall 1972 concluded that even minor variability in a range of climate parameters strongly influenced plague outbreaks in Vietnam. Their study would suggest that your analyses may simply be at too coarse a level to identify the true drivers.

Cavanaugh, D. C., & MARSHALL JR, J. D. (1972). The influence of climate on the seasonal prevalence of plague in the Republic of Vietnam. *Journal of Wildlife Diseases*, 8(1), 85-94.

9.) Indeed, even for Dengue, which has 100s of publications on climate and disease, there is active debate about how much including climate variables improves model fit or forecast accuracy. I would encourage the authors to consider a similar tact as was taken by Johansson et al. 2016 and ask whether including climate variables improves model fit beyond a seasonal model.

Johansson, M. A., Reich, N. G., Hota, A., Brownstein, J. S., & Santillana, M. (2016). Evaluating the performance of infectious disease forecasts: A comparison of climate-driven and seasonal dengue forecasts for Mexico. *Scientific reports*, 6, 33707.

10.) Additionally, a proposal from Schmid et al. suggested that climate-mediated, re-introductions of plague may have happened repeatedly from Asia to Europe. If so, could there

simply be a time effect due to the geographic nature of the introductions? E.g., if plague is introduced into N. Africa or S. Europe first, then all kinds of variables with N/S gradients would spuriously correlate with the timing of plague cases.

Schmid, B. V., Büntgen, U., Easterday, W. R., Ginzler, C., Walløe, L., Bramanti, B., & Stenseth, N. C. (2015). Climate-driven introduction of the Black Death and successive plague reintroductions into Europe. *Proceedings of the National Academy of Sciences*, 112(10), 3020-3025.

11.) Lastly, if the authors decide to pursue associations with climate variables, you might consider using published reconstructions. Although this is not my area of expertise.

Büntgen, U., Kyncl, T., Ginzler, C., Jacks, D. S., Esper, J., Tegel, W., ... & Kyncl, J. (2013). Filling the Eastern European gap in millennium-long temperature reconstructions. *Proceedings of the National Academy of Sciences*, 110(5), 1773-1778.

Decision letter (RSPB-2020-1407.R0)

07-Aug-2020

Dear Ms Krauer:

I am writing to inform you that your manuscript RSPB-2020-1407 entitled "The influence of temperature on the seasonality of historical plague outbreaks" has, in its current form, been rejected for publication in *Proceedings B*.

This action has been taken on the advice of referees, who have recommended that substantial revisions are necessary. With this in mind we would be happy to consider a resubmission, provided the comments of the referees are fully addressed. However please note that this is not a provisional acceptance.

Sincerely,
 Dr Sasha Dall
 mailto: proceedingsb@royalsociety.org

Associate Editor
 Board Member: 1

Comments to Author:

While the reviewers found the concept of the paper interesting and the data set compelling, both reviewers were uncompelled by the evidence presented for the main conclusions about the specific role that weather and climate play in modulating transmission. The reviewers provide helpful comments and recommended citations for more detail.

Reviewer(s)' Comments to Author:

Referee: 1

Comments to the Author(s)

This is an interesting study trying to document the seasonality of historic plague epidemics in Europe and Northern Africa. This work has the potential to add to our understanding of historic plague outbreaks, but the findings have to be discussed in context. The overall scientific argument has to be strengthened.

Major comments:

The abstract does not reflect the rest of the document well and should be rewritten to include justification of the study and implications of findings.

Please strengthen the justification for your study throughout your document i.e. the importance to understand historic epidemics to prepare for future epidemics. That something simply has not been done before is not enough to validate a study.

A major concern is the lack of background information on historic plague outbreaks in pre-industrial Europe. The authors need to provide a stronger argument for their hypothesis and a more nuanced discussion of their findings and the implications of their work. Literature suggests that the main driver for plague outbreaks in pre-industrial Europe and North-Africa was the maritime trading system (1). Indeed, the majority of the data on plague mortality records come from either port cities themselves, or from cities linked to the historic maritime trading system. Yet, the authors make a case for the seasonality of vector borne diseases such as plague mainly due to seasonal abundance of host and vectors. In the methods section the authors mention "introduction events" for the first time, but it is not clear if they refer to the absence of a reservoir or the zoonotic event. Indeed, studies suggest that there is no evidence for a plague reservoir in pre-industrial Europe (2), reducing the strength of a link between season, host and vector availability and plague even further. This may be the reason why model fit was so poor. However, linking the dynamics of plague outbreaks (growth rates) to climate variables, as has been done here, is sensible and in my opinion the findings of this part should be predominantly discussed. Not the occurrence of an outbreak, but its persistence and growth may be linked to environmental factors such as temperature, as it depends on the vector and host availability after the introductory event. The results on growth rate should be discussed in the context of a (temperature-modulated) trophic-cascade model for plague, including climatic effects on rodent abundance, flea abundance, and pathogen transmission.

It is not surprising that using average annual temperature and precipitation does not give clear results, as the plague transmission system relies on specific temperature and humidity ranges. Even though the lack of good data is mentioned in the limitations, the impact is not.

The discussion seems rushed and lacks clarity. The explanations of the studies limitations take up most of the space. It would improve flow if the argument was strengthened. Priority should be given to the findings on growth rates and their link to seasonality and temperature ranges. The difference between pneumonic and bubonic plague outbreaks in terms of sensitivity of transmission pathways to temperature should be clarified. Second pandemic outbreaks in Europe and northern Africa have been linked to multiple re-introductions via trade routes. Good climate conditions for vectors and hosts can positively affect growth rates after introduction.

Minor comments:

1. 22 Please add "in the northern hemisphere".

1. 41 please add: ..April and have been linked to temperature and precipitation (3, 4)
 1. 48 briefly describe the difference between modern day outbreaks and historic outbreaks in terms of transmission mechanisms. Endemic plague vs. introductory events on trade routes (1).
 1. 140 It is not clear why you used monthly plague mortality AND all-cause mortality in the same model? I assume you did not use both for the same location and outbreak.
 1. 218 please enter the exact p-values for the LR test until <0.001
 1. 222 change to "from 22 unique places"
 - 1.228 indicate the direction of the 1.4 week shift
- Fig 2. I suggest considering having calendar week on the x-axis. Here it can be seen that precipitation is not linearly related to epidemic calendar week and that should be added to the limitations of the study.
1. 352-358 The outbreak in Madagascar was mainly pneumonic as mentioned (5), strongly linked to human behaviour. The authors should add that most bubonic cases were recorded from the beginning of the usual plague season in Madagascar at the beginning of October when the climatic conditions are most favourable for vectors (6).

References

1. Yue RPH, Lee HF, Wu CYH. Trade routes and plague transmission in pre-industrial Europe. *Scientific Reports*. 2017;7(1):12973.
2. Schmid BV, Büntgen U, Easterday WR, Ginzler C, Walløe L, Bramanti B, et al. Climate-driven introduction of the Black Death and successive plague reintroductions into Europe. *Proceedings of the National Academy of Sciences*. 2015;112(10):3020.
3. Kreppel KS, Caminade C, Telfer S, Rajerison M, Rahalison L, Morse A, et al. A Non-Stationary Relationship between Global Climate Phenomena and Human Plague Incidence in Madagascar. *PLoS Negl Trop Dis*. 2014;8(10):e3155.
4. Stenseth N, Samia N, Viljugrein H, Kausrud K, Begon M, Davis S, et al. Plague dynamics are driven by climate variation. *Proc Natl Acad Sci USA*. 2006;103(35):13110 - 5.
5. Tsuzuki SaL, Hyojung and Miura, Fuminari and Chan, Yat Hin and Jung, Sung-mok and Akhmetzhanov, Andrei R and Nishiura, Hiroshi, . Dynamics of the pneumonic plague epidemic in Madagascar, August to October 2017. *Eurosurveillance*. 2017;22.
6. Kreppel KS, Telfer S, Rajerison M, Morse A, Baylis M. Effect of temperature and relative humidity on the development times and survival of *Synopsyllus fonquerniei* and *Xenopsylla cheopis*, the flea vectors of plague in Madagascar. *Parasites & Vectors*. 2016;9(1):82.

Referee: 2

Comments to the Author(s)

In their manuscript, Krauer et al. construct a large, historical data set on plague outbreaks and relate reported mortality to climate/weather variables. The authors have compiled and digitized an impressive set of data, which will undoubtedly facilitate quite a bit of future work on historical plague outbreaks. I also agree with the authors that studying the drivers of seasonality for infectious disease outbreaks remains an important and active area of research. While I found the seasonal differences between Algiers and Europe (including London)/ Alexandria intriguing, given the biased geographic nature of these data and the complex role precipitation and temperature play in the transmission of other pathogens, I remain skeptical of the conclusions regarding the specific role weather/climate plays in modulating transmission. Indeed, as a detail below, there seem to be many alternative hypotheses for how plague outbreaks are structured. I have the following, more specific comments, which I hope the authors find constructive.

- 1.) Do we have any sense for how these data might be biased due to general patterns of under-reporting and/or differences due to reporting practices by country? It may be that the signs of plague death are sufficiently telling that reporting is less of an issue than for other diseases, but the authors should clarify this point.

2.) Related to the above, one concern about peak timing is that this may be heavily influenced by reporting. For example, Park and Bolker recently discussed these issues from a theoretical perspective.

Park, S. W., & Bolker, B. M. (2020). A note on observation processes in epidemic models. *Bulletin of Mathematical Biology*, 82(3), 1-8.

3.) Additionally, to my knowledge, most climate-forced infectious disease models focus on changes in the effective number of secondary infections, R_e . The peak time may or may not change, depending on the population and household structure the disease is moving through, and other details of the assumed model. I would encourage the authors to focus on a different aspect of the epidemic curve or provide further justification for the use of peak timing.

Axelsen, J. B., Yaari, R., Grenfell, B. T., & Stone, L. (2014). Multiannual forecasting of seasonal influenza dynamics reveals climatic and evolutionary drivers. *Proceedings of the National Academy of Sciences*, 111(26), 9538-9542.

Volz, E. M., Miller, J. C., Galvani, A., & Meyers, L. A. (2011). Effects of heterogeneous and clustered contact patterns on infectious disease dynamics. *PLoS Comput Biol*, 7(6), e1002042.

4.) Can the authors expand a bit on how specifically their GEE model handles clustering and whether by clustering the authors mean having multiple observations from the same geographic location? It's more natural for me to think in terms of fixed effects from a mixed-effects model where location is a random effect or the central moment of the distribution describing a hyper parameter in a hierarchical Bayesian framework. I'm not necessarily advocating for a different model, just a bit more clarity.

5.) Did the authors consider leave-one-out cross validation performed at the country- or city-level where you both hold out individual outbreaks and entire locations? This might make a much more compelling case that there are general patterns related to temp/precip.

6.) Given all the myriad, climatic differences between these locations, e.g., Egypt, Algiers, Vienna, and London, how confident should we be that the associations with temp/precip aren't masking the true relationships? As an example, a paper on Whooping Cough outbreaks in London between 1700 and 1800 argued that climate variables were largely masking the effects of nutritional variation (or at least were only indirectly influencing disease dynamics).

Duncan, C. J., Duncan, S. R., & Scott, S. (1996). Whooping cough epidemics in London, 1701-1812: infection dynamics, seasonal forcing and the effects of malnutrition. *Proceedings of the Royal Society of London. Series B: Biological Sciences*, 263(1369), 445-450.

7.) For example, Ben Ari et al. 2011 argued that scale matters immensely when considering climate variables and modern plague dynamics.

Ari, T. B., Neerinx, S., Gage, K. L., Kreppel, K., LaDisoit, A., Leirs, H., & Stenseth, N. C. (2011). Plague and climate: scales matter. *PLoS Pathog*, 7(9), e1002160.

8.) Similarly, Cavanaugh & Marshall 1972 concluded that even minor variability in a range of climate parameters strongly influenced plague outbreaks in Vietnam. Their study would suggest that your analyses may simply be at too coarse a level to identify the true drivers.

Cavanaugh, D. C., & MARSHALL JR, J. D. (1972). The influence of climate on the seasonal prevalence of plague in the Republic of Vietnam. *Journal of Wildlife Diseases*, 8(1), 85-94.

9.) Indeed, even for Dengue, which has 100s of publications on climate and disease, there is active debate about how much including climate variables improves model fit or forecast accuracy. I

would encourage the authors to consider a similar tact as was taken by Johansson et al. 2016 and ask whether including climate variables improves model fit beyond a seasonal model.

Johansson, M. A., Reich, N. G., Hota, A., Brownstein, J. S., & Santillana, M. (2016). Evaluating the performance of infectious disease forecasts: A comparison of climate-driven and seasonal dengue forecasts for Mexico. *Scientific reports*, 6, 33707.

10.) Additionally, a proposal from Schmid et al. suggested that climate-mediated, re-introductions of plague may have happened repeatedly from Asia to Europe. If so, could there simply be a time effect due to the geographic nature of the introductions? E.g., if plague is introduced into N. Africa or S. Europe first, then all kinds of variables with N/S gradients would spuriously correlate with the timing of plague cases.

Schmid, B. V., Büntgen, U., Easterday, W. R., Ginzler, C., Walløe, L., Bramanti, B., & Stenseth, N. C. (2015). Climate-driven introduction of the Black Death and successive plague reintroductions into Europe. *Proceedings of the National Academy of Sciences*, 112(10), 3020-3025.

11.) Lastly, if the authors decide to pursue associations with climate variables, you might considering using published reconstructions. Although this is not my area of expertise.

Büntgen, U., Kyncl, T., Ginzler, C., Jacks, D. S., Esper, J., Tegel, W., ... & Kyncl, J. (2013). Filling the Eastern European gap in millennium-long temperature reconstructions. *Proceedings of the National Academy of Sciences*, 110(5), 1773-1778.

Author's Response to Decision Letter for (RSPB-2020-1407.R0)

See Appendix A.

RSPB-2020-2725.R0

Review form: Reviewer 1

Recommendation

Accept with minor revision (please list in comments)

Scientific importance: Is the manuscript an original and important contribution to its field?

Good

General interest: Is the paper of sufficient general interest?

Excellent

Quality of the paper: Is the overall quality of the paper suitable?

Good

Is the length of the paper justified?

Yes

Should the paper be seen by a specialist statistical reviewer?

Yes

Do you have any concerns about statistical analyses in this paper? If so, please specify them explicitly in your report.

No

It is a condition of publication that authors make their supporting data, code and materials available - either as supplementary material or hosted in an external repository. Please rate, if applicable, the supporting data on the following criteria.

Is it accessible?

Yes

Is it clear?

Yes

Is it adequate?

Yes

Do you have any ethical concerns with this paper?

No

Comments to the Author

Comments:

The authors have certainly addressed my concerns in a comprehensive manner and have explained/defended points they did not agree with well.

One of the remaining issues is the use of averages which is not followed up with conclusions. In the abstract and discussion, results on averages are presented that are meaningless.

In the second sentence of the discussion for example you write: "We show that on average, an outbreak lasted for 9 months, caused a total of 1352 deaths with 429 deaths at peak month, and infected 11% of the population."

Here, giving the reader more information about your data would be better. Replace that sentence with more meaningful values such as "...outbreak size varied from...to... months in length and caused between...and...deaths with the median number of deaths being..... Notably, on average 32% of deaths were recorded at the peak month.

The statement below from the abstract should be changed to the clearer language used in the discussion which conveys the key results much better.

You wrote: "We found that average annual temperature was negatively associated with peak timing: a decrease of the annual average temperature by 10°C delayed the peak timing by average 14 weeks. We further show that the predicted epidemic growth rates exhibit a non-linear relationship with temperature and are maximal at 17.3°C."

Change to: "We show that epidemic peak timing followed a latitudinal gradient, with average annual temperature negatively associated with peak timing. The predicted epidemic growth of all outbreaks was positive between 11.7°C and 21.5°C with a maximum at 17.3°C. Hence, our study provides evidence that the growth of plague epidemics across the whole study region depended on similar absolute temperature thresholds."

Minor comments:

Abstract:

P1 line 11 – add the word "...epidemic peak timing and growth rates of plague epidemics",

P1 line 15 - by an average of 14 weeks.

Discussion:

L 321 – This attack rate...

L 338 "seemed not associated" change to "did not seem associated".

L 391 "temperature data would allow taking into account

Review form: Reviewer 3

Recommendation

Major revision is needed (please make suggestions in comments)

Scientific importance: Is the manuscript an original and important contribution to its field?

Good

General interest: Is the paper of sufficient general interest?

Good

Quality of the paper: Is the overall quality of the paper suitable?

Acceptable

Is the length of the paper justified?

Yes

Should the paper be seen by a specialist statistical reviewer?

Yes

Do you have any concerns about statistical analyses in this paper? If so, please specify them explicitly in your report.

No

It is a condition of publication that authors make their supporting data, code and materials available - either as supplementary material or hosted in an external repository. Please rate, if applicable, the supporting data on the following criteria.

Is it accessible?

Yes

Is it clear?

Yes

Is it adequate?

Yes

Do you have any ethical concerns with this paper?

No

Comments to the Author

The article by Krauer et al entitled "The influence of temperature on the seasonality of historical plague outbreaks" investigates historical records describing plague epidemics during the Second Plague Pandemic across Europe and the its surrounding regions to address questions regarding the influencing factors affecting the seasonality of plague epidemics, mainly in relation to mean temperature and precipitation. In my view, the study is of interest and the paper is well written. Yet, there are a number of issues associated with the data interpretation and contextualisation that should be taken into account:

1. How were temperature estimates for each location retrieved and to what extent could the estimated peak of 17.3°C be influenced by differences between past and present annual temperature data? Does the identified temperature correlate with those reported during increased plague activity in areas that contain dense reservoirs today (for example in Central Asia)?
2. If the authors are not entirely confident on whether the presented temperature data can be an accurate reflection of the past, I would suggest that the absolute 17.3°C temperature value be

omitted from the paper's abstract. Instead, I would suggest presentation of a temperature range in association with epidemic increase and that the results be interpreted more generally in the context of seasonal changes.

3. The results of the study seem comparable with regions that currently sustain permanent plague reservoirs, or have sustained highly active reservoirs in the past (e.g. India and Argentina). In addition, it seems conceptually difficult to justify that temperature would have such a high influence on plague epidemics in the absence of a plague reservoir within the same region. Given these observations, the authors should discuss whether their results could add information to existing hypotheses regarding the existence of plague reservoirs in Europe during the Second Pandemic.

4. More specifically, recent aDNA data from plague epidemics of the Second Pandemic have shown a genetic continuum of *Y. pestis* strains isolated from western European outbreaks of the 14th-18th centuries. How are the present results interpreted alongside the growing evidence of plague persistence in western Europe during the Second Pandemic? Although the authors do mention that the existence of plague reservoirs in Europe is a debatable topic, the context of the debate is not described. Given the highly relevant topic of this paper and the potential impact of the presented results, a discussion of this new piece of evidence alongside other sources of data would be valuable.

5. Discussion Line 428: The example of plague in Madagascar in 2017 seems to be a poor one in this case, given the uncertainty shown around the number of confirmed versus probable cases for this outbreak, as well as the number of pneumonic versus bubonic cases. Therefore, I would suggest to omit this example from the discussion. For a recent study on the Madagascar outbreak please see:

Randremanana, R., et al., 2019. Epidemiological characteristics of an urban plague epidemic in Madagascar, August–November, 2017: an outbreak report. *The Lancet Infectious Diseases*, 19(5), pp.537-545.

6. Discussion Line 433: The example given about the 1630/31 Venice outbreak and its peak in October seems to contradict the paper's findings reporting epidemic peaks during spring in the Mediterranean. Yet, this example is presented here as being consistent with the paper's results. How is this discrepancy corroborated?

7. In the paper's abstract and introduction, the authors emphasize that a description of plague seasonality patterns in the past will be important for predicting future epidemics, though an explicit addressing of this hypothesis is not offered in the paper's discussion. How are the current results significant in the context of modern plague and what other contexts could the presented analytical methods/models be applicable to?

Decision letter (RSPB-2020-2725.R0)

29-Jan-2021

Dear Ms Krauer:

Your manuscript has now been peer reviewed and the reviews have been assessed by an Associate Editor. The reviewers' comments (not including confidential comments to the Editor) and the comments from the Associate Editor are included at the end of this email for your reference. As you will see, the reviewers and the Editors have raised some concerns with your manuscript and we would like to invite you to revise your manuscript to address them.

Research ethics:

Use of animals and field studies:

It is a condition of publication that you make available the data and research materials supporting the results in the article (<https://royalsociety.org/journals/authors/author-guidelines/#data>). Datasets should be deposited in an appropriate publicly available repository and details of the associated accession number, link or DOI to the datasets must be included in the Data Accessibility section of the article (<https://royalsociety.org/journals/ethics-policies/data-sharing-mining/>). Reference(s) to datasets should also be included in the reference list of the article with DOIs (where available).

Please submit a copy of your revised paper within three weeks. If we do not hear from you within this time your manuscript will be rejected. If you are unable to meet this deadline please let us know as soon as possible, as we may be able to grant a short extension.

Best wishes,
Dr Sasha Dall
mailto:proceedingsb@royalsociety.org

Associate Editor Board Member

Comments to Author:

Thank you for your re-submission.

Overall, the reviewers found the dataset compelling and the questions addressed interesting.

Many of the prior comments have been addressed - but some key reviewer questions remain. In particular, how are the temperature estimates generated, and how may they be influenced by differences between past and present annual temperature data? Further discussion of potential role of interannual variation and other climatic considerations is also still needed. Do these temperature estimates account for known climatic shifts over time period considered, such as the little ice ages?

Reviewer 3 notes the same concerns as one of the reviewers on the first submission about the relative importance of seasonality vs introductions, and the role of plague reservoirs. Although the authors have strengthened the discussion of the relative importance of these factors, it is clear that this issue requires even more consideration and discussion in the manuscript. Both reviewers provide several helpful suggestions to make the data descriptions in the text more comprehensive, and therefore more useful to readers, by focusing on ranges and variances rather than averages or exact values.

In addition reviewer 3 highlights that the abstract motivates the paper by future-casting from historical patterns, a promise that the manuscript does not deliver on. This hypothesis should be tested or removed.

Following revision which adequately addresses these concerns, this manuscript may be suitable for publication in Proc R Soc B.

Reviewer(s)' Comments to Author:

Referee: 1

Comments to the Author(s).

Comments:

The authors have certainly addressed my concerns in a comprehensive manner and have explained/defended points they did not agree with well.

One of the remaining issues is the use of averages which is not followed up with conclusions. In the abstract and discussion, results on averages are presented that are meaningless.

In the second sentence of the discussion for example you write: "We show that on average, an outbreak lasted for 9 months, caused a total of 1352 deaths with 429 deaths at peak month, and infected 11% of the population."

Here, giving the reader more information about your data would be better. Replace that sentence with more meaningful values such as "...outbreak size varied from...to... months in length and caused between ...and ...deaths with the median number of deaths being Notably, on average 32% of deaths were recorded at the peak month.

The statement below from the abstract should be changed to the clearer language used in the discussion which conveys the key results much better.

You wrote: "We found that average annual temperature was negatively associated with peak timing: a decrease of the annual average temperature by 10°C delayed the peak timing by average 14 weeks. We further show that the predicted epidemic growth rates exhibit a non-linear relationship with temperature and are maximal at 17.3°C."

Change to: "We show that epidemic peak timing followed a latitudinal gradient, with average annual temperature negatively associated with peak timing. The predicted epidemic growth of all outbreaks was positive between 11.7°C and 21.5°C with a maximum at 17.3°C. Hence, our study provides evidence that the growth of plague epidemics across the whole study region depended on similar absolute temperature thresholds."

Minor comments:

Abstract:

P1 line 11 - add the word "...epidemic peak timing and growth rates of plague epidemics",

P1 line 15 - by an average of 14 weeks.

Discussion:

L 321 - This attack rate...

L 338 "seemed not associated" change to "did not seem associated".

L 391 "temperature data would allow taking into account

Referee: 3

Comments to the Author(s).

The article by Krauer et al entitled "The influence of temperature on the seasonality of historical plague outbreaks" investigates historical records describing plague epidemics during the Second Plague Pandemic across Europe and the its surrounding regions to address questions regarding the influencing factors affecting the seasonality of plague epidemics, mainly in relation to mean temperature and precipitation. In my view, the study is of interest and the paper is well written. Yet, there are a number of issues associated with the data interpretation and contextualisation that should be taken into account:

1. How were temperature estimates for each location retrieved and to what extent could the estimated peak of 17.3°C be influenced by differences between past and present annual temperature data? Does the identified temperature correlate with those reported during increased plague activity in areas that contain dense reservoirs today (for example in Central Asia)?

2. If the authors are not entirely confident on whether the presented temperature data can be an accurate reflection of the past, I would suggest that the absolute 17.3°C temperature value be omitted from the paper's abstract. Instead, I would suggest presentation of a temperature range in association with epidemic increase and that the results be interpreted more generally in the context of seasonal changes.

3. The results of the study seem comparable with regions that currently sustain permanent plague reservoirs, or have sustained highly active reservoirs in the past (e.g. India and Argentina). In addition, it seems conceptually difficult to justify that temperature would have such a high influence on plague epidemics in the absence of a plague reservoir within the same region. Given these observations, the authors should discuss whether their results could add information to existing hypotheses regarding the existence of plague reservoirs in Europe during the Second Pandemic.

4. More specifically, recent aDNA data from plague epidemics of the Second Pandemic have shown a genetic continuum of *Y. pestis* strains isolated from western European outbreaks of the 14th-18th centuries. How are the present results interpreted alongside the growing evidence of plague persistence in western Europe during the Second Pandemic? Although the authors do mention that the existence of plague reservoirs in Europe is a debatable topic, the context of the debate is not described. Given the highly relevant topic of this paper and the potential impact of the presented results, a discussion of this new piece of evidence alongside other sources of data would be valuable.

5. Discussion Line 428: The example of plague in Madagascar in 2017 seems to be a poor one in this case, given the uncertainty shown around the number of confirmed versus probable cases for this outbreak, as well as the number of pneumonic versus bubonic cases. Therefore, I would suggest to omit this example from the discussion. For a recent study on the Madagascar outbreak please see:

Randremanana, R., et al., 2019. Epidemiological characteristics of an urban plague epidemic in Madagascar, August–November, 2017: an outbreak report. *The Lancet Infectious Diseases*, 19(5), pp.537-545.

6. Discussion Line 433: The example given about the 1630/31 Venice outbreak and its peak in October seems to contradict the paper's findings reporting epidemic peaks during spring in the Mediterranean. Yet, this example is presented here as being consistent with the paper's results. How is this discrepancy corroborated?

7. In the paper's abstract and introduction, the authors emphasize that a description of plague seasonality patterns in the past will be important for predicting future epidemics, though an explicit addressing of this hypothesis is not offered in the paper's discussion. How are the current results significant in the context of modern plague and what other contexts could the presented analytical methods/models be applicable to?

Author's Response to Decision Letter for (RSPB-2020-2725.R0)

See Appendix B.

RSPB-2020-2725.R1 (Revision)

Review form: Reviewer 1

Recommendation

Accept as is

Scientific importance: Is the manuscript an original and important contribution to its field?

Good

General interest: Is the paper of sufficient general interest?

Good

Quality of the paper: Is the overall quality of the paper suitable?

Good

Is the length of the paper justified?

Yes

Should the paper be seen by a specialist statistical reviewer?

Yes

Do you have any concerns about statistical analyses in this paper? If so, please specify them explicitly in your report.

No

It is a condition of publication that authors make their supporting data, code and materials available - either as supplementary material or hosted in an external repository. Please rate, if applicable, the supporting data on the following criteria.

Is it accessible?

Yes

Is it clear?

N/A

Is it adequate?

Yes

Do you have any ethical concerns with this paper?

No

Comments to the Author

I appreciate the revisions made by the authors in response to my concerns. The paper makes an important contribution to the field of plague research and, therefore, I recommend publication.

Decision letter (RSPB-2020-2725.R1)

04-Jun-2021

Dear Ms Krauer

I am pleased to inform you that your manuscript entitled "The influence of temperature on the seasonality of historical plague outbreaks" has been accepted for publication in Proceedings B.

Data Accessibility section

Open Access

Paper charges

Sincerely,

Dr Sasha Dall

Appendix A

We thank both referees for their useful comments and feedback, which has helped us to substantially improve our manuscript.

Overall, we have restructured and rewritten the introduction and the discussion to address the referees' points. We have also added a calculation of the attack rates (i.e. the proportion of the population infected at the end of the epidemic, also known as final size) for each epidemic (including a new figure), to provide some additional information, and to further illustrate the point of external forcing that leads to the decline of epidemics. Furthermore, we have added a new calculation of the distribution of cases by month and latitude, to better visualize our findings. We have also reworded some passages in methods and results to comply with the maximum word count. The conclusion from our study is unchanged, but we acknowledge and discuss alternative or additional factors that could have influenced the epidemic curves.

Please find our individual responses below.

Referee: 1

Comments to the Author(s)

This is an interesting study trying to document the seasonality of historic plague epidemics in Europe and Northern Africa. This work has the potential to add to our understanding of historic plague outbreaks, but the findings have to be discussed in context. The overall scientific argument has to be strengthened.

Major comments:

The abstract does not reflect the rest of the document well and should be rewritten to include justification of the study and implications of findings. Please strengthen the justification for your study throughout your document i.e. the importance to understand historic epidemics to prepare for future epidemics. That something simply has not been done before is not enough to validate a study.

We have extended the last part of the abstract to emphasize the implications of our study:

“These findings suggest that the pre-industrial outbreaks in our dataset were governed by the same underlying temperature thresholds. Our study provides a first systematic insight into the seasonality of historical plague in the northern hemisphere, as well as consistent evidence for a temperature-related process influencing the timing and growth of these epidemics. Given the importance of understanding past epidemics to prepare for future outbreaks, our study warrants further investigation into potential mechanisms.”

We have also added the following section in the introduction:

“Quantifying the seasonality of an infectious disease and its drivers may also provide insight into the mechanisms of historical plague in Europe. The existence of local reservoirs in pre-industrial Europe is controversial, and the vectors and hosts involved in disease transmission are not known [20]. Nowadays, the risk of another plague pandemic is fairly low due to the availability of antibiotics and reduced exposure to ectoparasites. Nevertheless, plague is still active in many parts of the world, and understanding the seasonality of historical outbreaks may contribute to understanding and preventing future epidemics.”

A major concern is the lack of background information on historic plague outbreaks in pre-industrial Europe. The authors need to provide a stronger argument for their hypothesis and a more nuanced discussion of their findings and the implications of their work. Literature suggests that the main driver for plague outbreaks in pre-industrial Europe and North-Africa was the

maritime trading system (1). Indeed, the majority of the data on plague mortality records come from either port cities themselves, or from cities linked to the historic maritime trading system. Yet, the authors make a case for the seasonality of vector borne diseases such as plague mainly due to seasonal abundance of host and vectors. In the methods section the authors mention “introduction events” for the first time, but it is not clear if they refer to the absence of a reservoir or the zoonotic event. Indeed, studies suggest that there is no evidence for a plague reservoir in pre-industrial Europe (2), reducing the strength of a link between season, host and vector availability and plague even further. This may be the reason why model fit was so poor.

We have restructured the discussion to address the issues raised here. We acknowledge the impact of trade in the long distance dissemination of plague. However, while trade activities were important for importing the disease in Europe, they had less influence on whether a local outbreak was successfully established or not. Not every introduction may have led to an epidemic. In fact, most of the disease introductions in European maritime ports during the third pandemic never led to large outbreaks [1]. All of the outbreaks in our dataset are the result of a successful establishment of an outbreak, many of which with hundreds of casualties. We also would like to stress that the peak timing is not simply a measure of whether an outbreak occurred or not. It is the moment when epidemic growth is zero, and thus likely determined by the same factors that influence the growth rate.

However, linking the dynamics of plague outbreaks (growth rates) to climate variables, as has been done here, is sensible and in my opinion the findings of this part should be predominantly discussed. Not the occurrence of an outbreak, but its persistence and growth may be linked to environmental factors such as temperature, as it depends on the vector and host availability after the introductory event. The results on growth rate should be discussed in the context of a (temperature-modulated) trophic-cascade model for plague, including climatic effects on rodent abundance, flea abundance, and pathogen transmission.

We have rewritten the introduction and the discussion where we discuss briefly the transmission cycle of bubonic plague in the traditional rodent-borne scenario and how meteorological factors relate to it. We have another, upcoming follow-up project for this paper, which further explores the ecological relationship of temperature and plague vectors. For this reason and to respect the maximum length restriction on the manuscript, we do not discuss the trophic cascade hypothesis here.

It is not surprising that using average annual temperature and precipitation does not give clear results, as the plague transmission system relies on specific temperature and humidity ranges. Even though the lack of good data is mentioned in the limitations, the impact is not.

We have restructured the discussion to clarify how the average annual temperature relates to the temperature threshold conducive of growth and thus to the peak timing. We show that the annual average temperature is strongly correlated with the time point when the minimum threshold for positive growth is attained (see Fig. S8). The results for the analysis of the peak timing also shows that the model including temperature explains the data better than a model just with an intercept (thus temperature is a significant predictor in the model). However, the variance explained by temperature is moderate, which suggests that temperature alone does not fully explain the relationship. We have thus rewritten the discussion to address other factors that influence the seasonality. Moreover, we have added a short section that discusses the negative results for precipitation:

“Precipitation on the other hand seemed not associated with neither peak timing nor epidemic growth. This finding contradicts ecological studies, which show that precipitation influences

plague incidence among rodents through effects on vegetation, fleas and hosts (trophic cascade hypothesis) [28]. Because the inter-annual variation is generally larger for precipitation than for temperature, true effects may be concealed due to the use of modern, averaged precipitation data. The relationship between precipitation and epidemic peak timing may also not be linear as suggested by few outliers, but it was not possible to test this with the present data. We did not observe an effect of precipitation in our analysis, but we cannot exclude that precipitation played a role in addition to temperature.”

The discussion seems rushed and lacks clarity. The explanations of the studies limitations take up most of the space. It would improve flow if the argument was strengthened. Priority should be given to the findings on growth rates and their link to seasonality and temperature ranges. The difference between pneumonic and bubonic plague outbreaks in terms of sensitivity of transmission pathways to temperature should be clarified. Second pandemic outbreaks in Europe and northern Africa have been linked to multiple re-introductions via trade routes. Good climate conditions for vectors and hosts can positively affect growth rates after introduction.

We have restructured and rewritten the discussion to make it more clear and allow readers to follow the arguments better. We hope you will find it improved and informative. The discussion of limitations is an important aspect of every study, and we have therefore kept in the paper. We have also extended the paragraph on the mechanisms that link temperature and plague. Moreover, we have added a new paragraph in the discussion on potential other explanations for the seasonality, which includes the role of introduction through trade routes.

Minor comments:

l. 22 Please add “in the northern hemisphere”. **done**

l. 41 please add: ...April and have been linked to temperature and precipitation (3, 4) **done**

l. 48 briefly describe the difference between modern day outbreaks and historic outbreaks in terms of transmission mechanisms. Endemic plague vs. introductory events on trade routes (1). **done**

l. 140 It is not clear why you used monthly plague mortality AND all-cause mortality in the same model? I assume you did not use both for the same location and outbreak. **We have added the word “further” to highlight that these are all-cause mortality data from other places. The purpose of this was to increase the sample size and explore the sensitivity of the analysis. We have not used both plague and all-cause mortality data from the same outbreak.**

l. 218 please enter the exact p-values for the LR test until <0.001 **done**

l. 222 change to “from 22 unique places” **done**

l.228 indicate the direction of the 1.4 week shift **we have replaced the word shift with “delayed”**

Fig 2. I suggest considering having calendar week on the x-axis. Here it can be seen that precipitation is not linearly related to epidemic calendar week and that should be added to the limitations of the study. **We follow the statistical convention to plot the explanatory variable on the x-axis and the outcome on the y-axis in order to correctly represent the direction of the relationship modelled.**

l. 352-358 The outbreak in Madagascar was mainly pneumonic as mentioned (5), strongly linked to human behaviour. The authors should add that most bubonic cases were recorded from the beginning of the usual plague season in Madagascar at the beginning of October when the climatic conditions are most favourable for vectors (6). **We have amended the sentence and added the suggested references.**

References

1. Yue RPH, Lee HF, Wu CYH. Trade routes and plague transmission in pre-industrial Europe. *Scientific Reports*. 2017;7(1):12973.
2. Schmid BV, Büntgen U, Easterday WR, Ginzler C, Walløe L, Bramanti B, et al. Climate-driven introduction of the Black Death and successive plague reintroductions into Europe. *Proceedings of the National Academy of Sciences*. 2015;112(10):3020.
3. Kreppel KS, Caminade C, Telfer S, Rajerison M, Rahalison L, Morse A, et al. A Non-Stationary Relationship between Global Climate Phenomena and Human Plague Incidence in Madagascar. *PLoS Negl Trop Dis*. 2014;8(10):e3155.
4. Stenseth N, Samia N, Viljugrein H, Kausrud K, Begon M, Davis S, et al. Plague dynamics are driven by climate variation. *Proc Natl Acad Sci USA*. 2006;103(35):13110 - 5.
5. Tsuzuki SaL, Hyojung and Miura, Fuminari and Chan, Yat Hin and Jung, Sung-mok and Akhmetzhanov, Andrei R and Nishiura, Hiroshi, . Dynamics of the pneumonic plague epidemic in Madagascar, August to October 2017. *Eurosurveillance*. 2017;22.
6. Kreppel KS, Telfer S, Rajerison M, Morse A, Baylis M. Effect of temperature and relative humidity on the development times and survival of *Synopsyllus fonquerniei* and *Xenopsylla cheopis*, the flea vectors of plague in Madagascar. *Parasites & Vectors*. 2016;9(1):82.

Referee: 2

Comments to the Author(s)

In their manuscript, Krauer et al. construct a large, historical data set on plague outbreaks and relate reported mortality to climate/weather variables. The authors have compiled and digitized an impressive set of data, which will undoubtedly facilitate quite a bit of future work on historical plague outbreaks. I also agree with the authors that studying the drivers of seasonality for infectious disease outbreaks remains an important and active area of research. While I found the seasonal differences between Algiers and Europe (including London)/Alexandria intriguing, given the biased geographic nature of these data and the complex role precipitation and temperature play in the transmission of other pathogens, I remain skeptical of the conclusions regarding the specific role weather/climate plays in modulating transmission. Indeed, as a detail below, there seem to be many alternative hypotheses for how plague outbreaks are structured. I have the following, more specific comments, which I hope the authors find constructive.

We appreciate the acknowledgment of the extensive plague dataset that has been collected and digitized in the present study. In the new version of the manuscript, we have provided a better overview of alternative hypotheses for how plague outbreaks are structured, and at the same time strengthened our argument for the hypothesis of historical plague outbreaks showing a seasonal trend and the seasonality of plague being related to temperature thresholds.

1.) Do we have any sense for how these data might be biased due to general patterns of under-reporting and/or differences due to reporting practices by country? It may be that the signs of plague death are sufficiently telling that reporting is less of an issue than for other diseases, but the authors should clarify this point.

Given the nature of historical data, we never have full confidence in what the data represent. This is a general limitation of historical epidemiology, which is very difficult to address given how little we know about reporting. Generally, we believe that deaths are less likely to be overlooked than living cases, and under- (or overreporting) may have been a smaller issue. There are few

estimates for underreporting in historical sources [2]. We address this issue in an upcoming paper, where we specifically estimate the underreporting in London from plague and all-cause mortality data. In this paper, we have rewritten the discussion of the limitations and briefly discuss the effects of underreporting on the outcomes we measure in this analysis.

2.) Related to the above, one concern about peak timing is that this may be heavily influenced by reporting. For example, Park and Bolker recently discussed these issues from a theoretical perspective.

Park, S. W., & Bolker, B. M. (2020). A note on observation processes in epidemic models. *Bulletin of Mathematical Biology*, 82(3), 1-8.

We acknowledge that delayed reporting can influence the reconstruction of the epidemiological peak. However, this is mostly a problem for the data collection and evaluation of ongoing outbreaks. We assume that the date of death of a family member was usually well known both to relatives and church members, because this information was often meticulously recorded in parish burial register. Religious traditions and hygiene considerations must have also required deaths to be buried as soon as possible, which makes it rather unlikely that there were large delays in acknowledging deaths.

3.) Additionally, to my knowledge, most climate-forced infectious disease models focus on changes in the effective number of secondary infections, R_e . The peak time may or may not change, depending on the population and household structure the disease is moving through, and other details of the assumed model. I would encourage the authors to focus on a different aspect of the epidemic curve or provide further justification for the use of peak timing..

Axelsen, J. B., Yaari, R., Grenfell, B. T., & Stone, L. (2014). Multiannual forecasting of seasonal influenza dynamics reveals climatic and evolutionary drivers. *Proceedings of the National Academy of Sciences*, 111(26), 9538-9542.

Volz, E. M., Miller, J. C., Galvani, A., & Meyers, L. A. (2011). Effects of heterogeneous and clustered contact patterns on infectious disease dynamics. *PLoS Comput Biol*, 7(6), e1002042.

In contrast to R_e , the growth rate of an epidemic makes no assumption about the mode of transmission nor the serial interval. It is thus a better measure of epidemic growth when circumstances of transmission are unclear. The epidemic growth model also makes no assumption about homogeneous or heterogeneous contact structures. These aspects are usually modelled with an SIR-type transmission model. The growth rate on the other hand is a phenomenological model.

4.) Can the authors expand a bit on how specifically their GEE model handles clustering and whether by clustering the authors mean having multiple observations from the same geographic location? It's more natural for me to think in terms of fixed effects from a mixed-effects model where location is a random effect or the central moment of the distribution describing a hyper parameter in a hierarchical Bayesian framework. I'm not necessarily advocating for a different model, just a bit more clarity.

The GEE model is a common approach for modelling clustered data, when the interest is to adjust for the clustering and uncover the population average, but not specifically to quantify the effect for each observation in the cluster. Of note, a mixed effects model yielded almost the

same estimates, but we here preferred the simpler approach of the GEE model.

5.) Did the authors consider leave-one-out cross validation performed at the country- or city-level where you both hold out individual outbreaks and entire locations? This might make a much more compelling case that there are general patterns related to temp/precip.

Cross validation usually requires a large enough sample size, which in our case was not sufficient at country or city level. However, we have investigated the sensitivity of the analysis of temperature and epidemic peak timing by adding additional data for further outbreaks, which did not change our conclusions.

6.) Given all the myriad, climatic differences between these locations, e.g., Egypt, Algiers, Vienna, and London, how confident should we be that the associations with temp/precip aren't masking the true relationships? As an example, a paper on Whooping Cough outbreaks in London between 1700 and 1800 argued that climate variables were largely masking the effects of nutritional variation (or at least were only indirectly influencing disease dynamics).

Duncan, C. J., Duncan, S. R., & Scott, S. (1996). Whooping cough epidemics in London, 1701-1812: infection dynamics, seasonal forcing and the effects of malnutrition. Proceedings of the Royal Society of London. Series B: Biological Sciences, 263(1369), 445-450.

We have restructured the discussion and address potential other explanations.

7.) For example, Ben Ari et al. 2011 argued that scale matters immensely when considering climate variables and modern plague dynamics.

We agree that climate affects plague at different spatial and temporal levels and that both seasonal trends and inter-annual variations are important. However, as we already mention in the manuscript, there are no long-term temperature data available for all these places going back as early as the 14th century. We acknowledge in the discussion, that inter-annual variation could potentially influence peak timing and growth for a given place.

Ari, T. B., Neerinckx, S., Gage, K. L., Kreppel, K., LaDiso, A., Leirs, H., & Stenseth, N. C. (2011). Plague and climate: scales matter. PLoS Pathog, 7(9), e1002160.

8.) Similarly, Cavanaugh & Marshall 1972 concluded that even minor variability in a range of climate parameters strongly influenced plague outbreaks in Vietnam. Their study would suggest that your analyses may simply be at too coarse a level to identify the true drivers.

Cavanaugh, D. C., & MARSHALL JR, J. D. (1972). The influence of climate on the seasonal prevalence of plague in the Republic of Vietnam. Journal of Wildlife Diseases, 8(1), 85-94.

The aim of this study was not to identify all the true drivers of seasonality or to quantify long-term trends and inter-annual variations. The aim of this study was to specifically quantify the effect of temperature and precipitation on the seasonal pattern. We hope the rewritten discussion conveys this message.

9.) Indeed, even for Dengue, which has 100s of publications on climate and disease, there is active debate about how much including climate variables improves model fit or forecast accuracy. I would encourage the authors to consider a similar tact as was taken by Johansson

et al. 2016 and ask whether including climate variables improves model fit beyond a seasonal model.

Johansson, M. A., Reich, N. G., Hota, A., Brownstein, J. S., & Santillana, M. (2016). Evaluating the performance of infectious disease forecasts: A comparison of climate-driven and seasonal dengue forecasts for Mexico. *Scientific reports*, 6, 33707.

As we mention in the discussion, unfortunately there are no contemporary data on short- or long-term climate for any of the outbreaks (with the exception of Cairo, see supplement). Such an analysis would be very interesting, but is unfortunately not possible.

10.) Additionally, a proposal from Schmid et al. suggested that climate-mediated, re-introductions of plague may have happened repeatedly from Asia to Europe. If so, could there simply be a time effect due to the geographic nature of the introductions? E.g., if plague is introduced into N. Africa or S. Europe first, then all kinds of variables with N/S gradients would spuriously correlate with the timing of plague cases.

Schmid, B. V., Büntgen, U., Easterday, W. R., Ginzler, C., Walløe, L., Bramanti, B., & Stenseth, N. C. (2015). Climate-driven introduction of the Black Death and successive plague reintroductions into Europe. *Proceedings of the National Academy of Sciences*, 112(10), 3020-3025.

We have rewritten the discussion to address the issue of re-introduction vs. successful establishment of an outbreak.

11.) Lastly, if the authors decide to pursue associations with climate variables, you might consider using published reconstructions. Although this is not my area of expertise.

Büntgen, U., Kyncl, T., Ginzler, C., Jacks, D. S., Esper, J., Tegel, W., ... & Kyncl, J. (2013). Filling the Eastern European gap in millennium-long temperature reconstructions. *Proceedings of the National Academy of Sciences*, 110(5), 1773-1778.

This is an interesting paper, but it is beyond the scope of this study to explore the effect of inter-annual variation based on model predictions of historical temperature. Moreover, the precision of these model-based predictions may not be sufficient to capture the local, inter-annual variation.

1. Bramanti B, Dean KR, Walloe L, Chr Stenseth N. The Third Plague Pandemic in Europe. *Proc Biol Sci*. 2019;286(1901):20182429. doi:10.1098/rspb.2018.2429.
2. Graunt J. London's dreadful visitation: or, A collection of all the bills of mortality for this present year: beginning the 27th of December 1664, and ending the 19th of December following: as also, the general or whole years bill: according to the report made to the King's most excellent Majesty. London: Cotes; 1665.

Appendix B

Referee 1:

The authors have certainly addressed my concerns in a comprehensive manner and have explained/defended points they did not agree with well.

We would like to thank Referee 1 for their thoughtful and detailed comments.

One of the remaining issues is the use of averages which is not followed up with conclusions. In the abstract and discussion, results on averages are presented that are meaningless. In the second sentence of the discussion for example you write: "We show that on average, an outbreak lasted for 9 months, caused a total of 1352 deaths with 429 deaths at peak month, and infected 11% of the population." Here, giving the reader more information about your data would be better. Replace that sentence with more meaningful values such as "...outbreak size varied from....to ... months in length and caused between ...and ...deaths with the median number of deaths being Notably, on average 32% of deaths were recorded at the peak month.

We have removed the corresponding numbers in the discussion as it was a mere repetition of the results. In the results section, we provide the summary estimates including an average point estimate and a range.

The statement below from the abstract should be changed to the clearer language used in the discussion which conveys the key results much better.

You wrote: "We found that average annual temperature was negatively associated with peak timing: a decrease of the annual average temperature by 10°C delayed the peak timing by average 14 weeks. We further show that the predicted epidemic growth rates exhibit a non-linear relationship with temperature and are maximal at 17.3°C."

Change to: "We show that epidemic peak timing followed a latitudinal gradient, with average annual temperature negatively associated with peak timing. The predicted epidemic growth of all outbreaks was positive between 11.7°C and 21.5°C with a maximum at 17.3°C. Hence, our study provides evidence that the growth of plague epidemics across the whole study region depended on similar absolute temperature thresholds."

Done

Minor comments:

Abstract:

P1 line 11 – add the word "...epidemic peak timing and growth rates of plague epidemics",

P1 line 15 - by an average of 14 weeks.

Discussion:

L 321 – This attack rate...

L 338 "seemed not associated" change to "did not seem associated".

L 391 "temperature data would allow taking into account

Referee 3:

The article by Krauer et al entitled "The influence of temperature on the seasonality of historical plague outbreaks" investigates historical records describing plague epidemics during the Second Plague Pandemic across Europe and the its surrounding regions to address questions regarding the influencing factors affecting the seasonality of plague epidemics, mainly in relation to mean temperature and precipitation. In my view, the study is of interest and the paper is well written.

We would like to thank Referee 3 for their thoughtful and detailed comments.

Yet, there are a number of issues associated with the data interpretation and contextualisation that should be taken into account:

1. How were temperature estimates for each location retrieved and to what extent could the estimated peak of 17.3°C be influenced by differences between past and present annual temperature data? Does the identified temperature correlate with those reported during increased plague activity in areas that contain dense reservoirs today (for example in Central Asia)?

As described in the methods section, the temperature data is from the CRU TS dataset that includes monthly averaged temperatures as a raster file. We extracted these data for each point location for the years 1901 to 1939. We estimated that the maximum peak growth occurred around 17.3°C based on the modern temperature data. It is difficult to speculate on the how much this temperature value could change if we had contemporaneous meteorological data. We have done our best to address this issue in the supplementary text S2, using the available data from the 1835 outbreak of plague in Cairo. For this outbreak, the difference in temperatures between the modern and historical data could be up to 1.5°C. However, we also cannot attest to the accuracy of temperature readings from the 1830s. We have rewritten the discussion to more clearly address the limitations of our data:

The use of modern meteorological data is a limitation of our study. As we show in the supplemental material, historical temperatures may have differed from modern temperatures (Text S2, Figure S9). The second pandemic occurred during a prolonged period of cooling in Europe known as the Little Ice Age. Thus, the temperature range conducive of epidemic growth could have been marginally lower than our estimates. The availability of only averaged modern data also meant that we could not consider inter-annual differences in the seasonal temperature and precipitation, or their lagged effects. Our results should thus be considered an approximation of the general meteorological process at a seasonal scale. Contemporaneous meteorological data may have also explained more of the variance in the peak timing and growth rates, particularly for precipitation, which can exhibit large variation between years."

The agreement of the identified favourable temperature range with modern data is addressed in a rewritten paragraph in the discussion:

"In the Malagasy highlands, outbreaks in the last decades commonly grow rapidly in August and September [76] when the average temperature is around 15–20°C, which is in the same range as predicted by our model. However, not all plague outbreaks may show the same temperature thresholds we found in our

analysis. Outbreaks in Bombay usually increased in December to January and peaked in February to April [77]. The average temperature in Bombay drops below 25°C in December and January and rises again above 25°C in March. Similarly, in Uganda, outbreaks increase typically in August to October during a period of high precipitation when the temperature is around 22°C [78]."

There may be further examples, where our prediction is or is not in agreement with data from other outbreaks, but a systematic analysis of the available data (as we have done here) would be necessary to address this thoroughly. However, this is beyond the scope of this paper. For this reason, and to respect the word count, we have not elaborated further on this topic.

2. If the authors are not entirely confident on whether the presented temperature data can be an accurate reflection of the past, I would suggest that the absolute 17.3°C temperature value be omitted from the paper's abstract. Instead, I would suggest presentation of a temperature range in association with epidemic increase and that the results be interpreted more generally in the context of seasonal changes.

We believe that despite the shortcomings of modern temperature data, the mean (point estimate) is still useful as an approximation for the maximum growth period, and we have decided to keep the information. We have rephrased the sentence:

"Based on modern temperature data, the predicted epidemic growth of all outbreaks was positive between 11.7°C and 21.5°C with a maximum around 17.3°C"

We have also added the following sentence in the discussion of the limitations:

"Our results should thus be considered an approximation of the general meteorological process at a seasonal scale."

3. The results of the study seem comparable with regions that currently sustain permanent plague reservoirs, or have sustained highly active reservoirs in the past (e.g. India and Argentina). In addition, it seems conceptually difficult to justify that temperature would have such a high influence on plague epidemics in the absence of a plague reservoir within the same region. Given these observations, the authors should discuss whether their results could add information to existing hypotheses regarding the existence of plague reservoirs in Europe during the Second Pandemic.

The debate of the reservoir and the modes of transmission in second pandemic Europe is important and interesting, but to respect the word count of the manuscript, we have kept the discussion in this manuscript short. We have rewritten the paragraph about potential mechanisms (including reservoirs) as follows:

*"There is an ongoing debate about which plague hosts and vectors were present in Europe during the second pandemic and whether the disease was maintained in local reservoirs (see e.g. [66, 67]). The establishment of a persistent reservoir in the Alpine region of Western Europe has been proposed [68]. However, recent studies of ancient DNA suggest that the genetic diversity of *Y. pestis* in Europe may be the result of multiple introductions [15] possibly from multiple reservoirs located in western Eurasia [69]."*

Since meteorological conditions can influence plague transmission through several pathways, the association in our study between temperature and the seasonal development of plague epidemics alone cannot provide evidence for or against a rodent reservoir in Europe. It is also unclear to what extent direct transmission routes (pneumonic or through human ectoparasites) played a role [67]. The contribution of climatic conditions to pneumonic transmission has not been addressed systematically. However, the occurrence of pneumonic outbreaks in many climatically diverse places such as Manchuria in December to January ([70], average temperature around -10°C) or Rangoon (Myanmar) in September ([71], average temperature 25-30°C) suggests that the suitable temperature range may be very large, and that other factors such as crowding and the influx of susceptibles may be more influential. The influence of temperature on human ectoparasites and the potential effect on plague in humans has not been studied so far and thus, we cannot evaluate it in comparison to our results.”

4. More specifically, recent aDNA data from plague epidemics of the Second Pandemic have shown a genetic continuum of *Y. pestis* strains isolated from western European outbreaks of the 14th-18th centuries. How are the present results interpreted alongside the growing evidence of plague persistence in western Europe during the Second Pandemic? Although the authors do mention that the existence of plague reservoirs in Europe is a debatable topic, the context of the debate is not described. Given the highly relevant topic of this paper and the potential impact of the presented results, a discussion of this new piece of evidence alongside other sources of data would be valuable.

See reply above

5. Discussion Line 428: The example of plague in Madagascar in 2017 seems to be a poor one in this case, given the uncertainty shown around the number of confirmed versus probable cases for this outbreak, as well as the number of pneumonic versus bubonic cases. Therefore, I would suggest to omit this example from the discussion. For a recent study on the Madagascar outbreak please see:
Randremanana, R., et al., 2019. Epidemiological characteristics of an urban plague epidemic in Madagascar, August–November, 2017: an outbreak report. *The Lancet Infectious Diseases*, 19(5), pp.537-545.

The seasonality of plague in Madagascar is very consistent also for many years prior to the last large outbreak. Some of these outbreaks may have included pneumonic cases, but this does not invalidate the comparison. It is likely that also historical outbreaks may have consisted of both bubonic and pneumonic cases. We have therefore rephrased the sentence and added a better reference:

“In the Malagasy highlands, outbreaks commonly grow rapidly in August and September [13] when the average temperature is around 15-20°C, which is in the same range as predicted by our model.”

6. Discussion Line 433: The example given about the 1630/31 Venice outbreak and its peak in October seems to contradict the paper’s findings reporting epidemic peaks during spring in the Mediterranean.

Yet, this example is presented here as being consistent with the paper's results. How is this discrepancy corroborated?

The peak timing in spring is restricted to places in the Southern Mediterranean and Africa. Venice is quite far up North and its average temperature is only 12.6°C. The climatically closest place in our dataset is Florence (12.06°). The estimated peak weeks for outbreaks in Florence are weeks 38-44, thus the peak in Venice is late in the year but still within the range of the data.

7. In the paper's abstract and introduction, the authors emphasize that a description of plague seasonality patterns in the past will be important for predicting future epidemics, though an explicit addressing of this hypothesis is not offered in the paper's discussion. How are the current results significant in the context of modern plague and what other contexts could the presented analytical methods/models be applicable to?

We have removed mentions of the applicability to future epidemics from the abstract and introduction.